



# Real-time UV-Index retrieval in Europe using Earth Observation based techniques and validation against ground-based measurements

Panagiotis G. Kosmopoulos[1], Stelios Kazadzis[2], Alois W. Schmalwieser[3], Panagiotis I. Raptis[1], Kyriakoula Papachristopoulou[1], Ilias Fountoulakis[1,4], Akriti Masoom[5], Alkiviadis F. Bais[6], Julia Bilbao[7], Mario Blumthaler[8], Axel Kreuter[8,9], Anna Maria Siani[10], Kostas Eleftheratos[11], Chrysanthi Topaloglou[5], Julian Gröbner[2], Bjørn Johnsen[12], Tove Svendby[13], Jose Manuel Vilaplana[14], Lionel Doppler[15], Ann R. Webb[16], Marina Khazova[17], Hugo De Backer[18], Anu Heikkilä[19], Kaisa Lakkala[20], Janusz Jaroslawski[21], Charikleia Meleti[5], Henri Diémoz[4], Gregor Hülsen[2], Barbara Klotz[8], John Rimmer[16], Charalampos Kontoes[1]

[1]National Observatory of Athens, Athens, Greece
[2]Physikalisch-Meteorologisches Observatorium Davos, World Radiation Center, Davos Dorf, Switzerland
[3]University of Veterinary Medicine, Vienna, Austria
[4]Environmental Protection Agency of Aosta Valley, Aosta, Italy
[5]Indian Institute of Technology Roorkee, India
[6]Aristotle University of Thessaloniki, Greece
[7]Valladolid University, Valladolid, Spain
[8]Innsbruck Medical University, Innsbruck, Austria
[9] Luftblick OG, Innsbruck, Austria
[10]Sapienza University of Rome, Rome, Italy
[11]National and Kapodistrian University of Athens, Athens, Greece
[12]Norwegian Radiation and Nuclear Safety Authority, Bærum, Norway
[13]Norwegian Institute for Air Research, Bærum, Norway
[14]National Institute for Aerospace Technology, Torrejón de Ardoz, Spain
[15]German Weather Service, Offenbach, Germany
[16]University of Manchester, Manchester, UK
[17]Public Health England, London, UK
[18]Royal Meteorological Institute of Belgium, Bruxelles, Belgium
[19]Finnish Meteorological Institute, Helsinki, Finland
[20]Finnish Meteorological Institute, Sodankylä, Finland
[21]Institute of Geophysics, Polish Academy of Sciences, Warsaw, Poland

*Correspondence to:* P.G. Kosmopoulos (pkosmo@noa.gr)

**Abstract.** This study introduces an Earth observation (EO)-based system which is capable of operationally estimating and continuously monitoring the ultraviolet index (UVI) in Europe. The UVIOS (i.e. UV-Index Operating System) exploits a synergy of radiative transfer models with high performance computing and EO data from satellites (Meteosat Second Generation and Meteorological Operational Satellite-B), and retrieval processes (Tropospheric Emission Monitoring Internet Service, Copernicus Atmosphere Monitoring Service and the Global Land Service). It provides a near-real-time now-casting





and short-term forecasting service for UV radiation over Europe. The main atmospheric inputs for the UVI simulations include
ozone, clouds and aerosols while the impacts of ground elevation and surface albedo are also taken into account. The UVIOS
output is the UVI at high spatial and temporal resolution (5 km and 15 minutes, respectively) for Europe (i.e. 1.5 million pixels)
in real-time. The UVI is empirically related to biologically important UV dose rates and the reliability of this EO-based solution
was verified against ground-based measurements from 17 stations across Europe. Stations are equipped with spectral,
broadband or multi-filter instruments and cover a range of topographic and atmospheric conditions. A period of over one year
of forecasted 15-min retrievals under all sky conditions were compared with the ground–based measurements. UVIOS
forecasts were within ±0.5 of measured UVI for at least 70% of the data compared at all stations. For clear sky conditions the
agreement was better than 0.5 UVI for 80% of the data. A sensitivity analysis of EO inputs and UVIOS outputs was performed
in order to quantify the level of uncertainty in the derived products, and to identify the covariance between the accuracy of the
output and the spatial and temporal resolution, and the quality of the inputs. Overall, UVIOS slightly overestimated UVI due
to observational uncertainties in inputs of cloud and aerosol. This service will hopefully contribute to EO capabilities and will
assist the provision of operational early warning systems that will help raise awareness among European Union citizens of the
health implications of high UVI doses.
**Keywords.** Ultraviolet Index; Earth Observation; Radiative Transfer; High Performance Computing; Clouds; Aerosols;
Ozone; Solar Zenith Angle; Ground Elevation; Surface Albedo
**1 Introduction**
Human exposure to ultraviolet (UV) radiation has both beneficial and harmful effects (Juzeniene et al., 2011; Lucas et al.,
2006). Overexposure to UV radiation (UVR) has a number of implications, such as the acute response of erythema, the risk of
skin cancer and a number of eye diseases (snow blindness, cataract). Nevertheless, exposure to solar UVB radiation is the main
mechanism for the synthesis of Vitamin D in the human skin (Holick, 2002; Webb and Engelsen, 2008; Webb et al., 2011).
Low levels of the Vitamin D are associated with depression of the immune system and there is evidence that is linked to a
number of medical implications (Lucas et al., 2015).
The UV index was introduced by WHO/WMO in 1994 (WMO, 1995), as a simple method of informing the general public
about the erythema effective (sun-burning) UV. It is a unitless, scaled version of erythemally-weighted UV determined by
multiplying the erythema weighted irradiance (in $W/m^2$) by 40 $m^2/W$ (Fioletov et al., 2010 ; Vanicek et al., 2000; WHO, 2002).
The response of UV radiation to climatic changes is of great concern (Bais et al., 2019; Bais et al., 2018; McKenzie et al.,
2011). According to the latest work of Bais et al. (2019) greater values of UV are expected by the end of 21st century, relative
to the present decade, at low latitudes,  while at higher latitudes UV will decrease  but these projections are associated with
high uncertainty (up to 30%).



There are many factors affecting UV irradiance reaching Earth's surface (Kerr and Fioletov, 2008). The dependence of UV
irradiance on astronomical and geometrical parameters is generally well understood, and in many cases the changes are
periodical (e.g. (Blumthaler et al., 1997; Gröbner et al., 2017; Larkin et al., 2000; Seckmeyer et al., 2008)). Atmospheric gases
play a crucial role in attenuating UV irradiance, with $O_3$ being the main absorber in the UVB (Bais et al., 1993), and $NO_2$ being
the main absorber in the UV-A (e.g. Cede et al. (2006)) spectral region. Aerosols are another important parameter controlling
UV irradiance levels at the surface (e.g. Kazadzis et al. (2009b)). Aerosol optical depth (AOD) that quantifies the attenuation
of the direct solar beam by aerosols is a parameter varying with wavelength, as well as single scattering albedo (SSA), which
determines the scattering ratio to total extinction. Several recent studies based on incoming UV irradiance measurements or
calculations reveal the enhanced absorption by aerosols in the UV relative to the visible spectral range. They show the
importance of using SSA in the UV spectral region, instead of interpolating SSA at visible wavelengths to the UV, or directly
using SSA at visible wavelengths, options that systematically overestimate UV irradiance (Corr et al., 2009; Fountoulakis et
al., 2019; Kazadzis et al., 2016; Mok et al., 2018; Raptis et al., 2018).
All the aforementioned parameters are particularly important under cloud free conditions. The cloudy sky complicates the
propagation of solar radiation, predominantly in the troposphere, through multiple cloud - radiation interactions. Nonetheless,
UVR is less affected than the total solar radiation by clouds (e.g. Badosa et al. (2014)). Bais et al. (1993) quantified that for
the city of Thessaloniki the change from 0 to 8/8 for cloud coverage corresponds to 80% reduction in the UVR and pointed
out that there is very low wavelength dependence  of UVR attenuation by cloud cover. In other studies (Mayer et al., 1998;
Seckmeyer et al., 1996) the authors showed that although the transmittance of clouds does not vary significantly with
wavelength in the UV, clouds give a wavelength dependent effect to the diffuse component the surface UVR, due to the more
effective scattering and absorption of shorter UV wavelengths in the atmosphere acting on larger air masses.  In cases of
partially cloudy sky but unobscured sun, UVR tends to be higher than in clear sky conditions (e.g. Badosa et al. (2014)), as is
the case for total solar radiation. For short timescale analysis the variability of UVR introduced by clouds should be considered.
A review of empirical studies of cloud effects on UVR is given by Calbó et al. (2005).
Solar UV irradiance at the surface increases with increasing surface albedo. This increment affects the UV radiant exposure,
which becomes crucial for outdoor human activities (Schmalwieser and Siani, 2018; Schmalwieser, 2020; Siani et al., 2008).
Measurements and computations of effective surface albedo for heterogeneous surfaces reveal its strong spectral dependence,
with snow-covered surfaces having significantly higher values of albedo for short wavelengths compared to total solar radiation
(Blumthaler and Ambach, 1988; Kreuter et al., 2014). Stronger enhancement of the UV relative to visible radiation over highly
reflective surfaces is also due to the more effective multiple scattering of shorter wavelengths in the atmosphere.
Any systematic changes in any of the parameters described in previous paragraphs have the potential to lead to changes for
UVR. These changes vary significantly throughout the globe and are attributed to different possible drivers (Bernhard and
Stierle, 2020; Fountoulakis et al., 2018; McKenzie et al., 2019). Fountoulakis et al. (2020a) gives a review of recent
publications concerning UV trends since 1990s, and associated factors, summarizing these as positive trends for South and
Central Europe and negative trends at higher latitudes, and recognizing the important role of aerosols and cloud coverage for



these trends in contrast to ozone. Findings from the same study demonstrated that the long – term changes of UV irradiance
recorded at four stations around Europe during the last two decades are mainly attributed to aerosols, cloud coverage and
surface albedo variations, with total ozone changes being of minor importance.
For the northern mid-latitudes Zerefos et al. (2012) showed that the long-term (1995-2006) positive trends in total ozone wasn't
enough to compensate for, let alone reverse, the UVB increase attributed to tropospheric aerosol decline (brightening effect).
Since 2007, a slowdown or even a possible turning point in the positive UVB trend was detected, which was attributed to the
continued upward trend in total ozone overwhelming the aerosol effect.By contrast, the long-term variability of UVB irradiance
over northern high latitudes was determined by ozone and not by aerosol trends, as shown by Eleftheratos et al. (2015) who
found a statistically significant negative trend of -3.9% per decade for the UVB irradiance during the time period 1999-2011,
from Ground-Based  (GB) measurements at 7 stations. This was in agreement with statistically significant increase of
spaceborne measured total ozone by about 1.5% per decade (ozone recovery) for the same area. For Arctic regions changes in
snow cover have a great impact on UV trends according to Bernhard (2011), who concluded that the future Arctic UV climate
may be affected more by a warming climate changing the snow cover than changes in stratospheric ozone concentrations.
The continuous monitoring of the UV index is currently performed by about 160 stations from 25 countries around Europe
(Schmalwieser et al., 2017), with all monitoring instruments having the potential to provide other effective doses such as the
effective dose for the production of vitamin D in human skin (e.g. Fioletov et al. (2009)).
There are three types of instruments for UV irradiance measurements; those measuring the integral of UV irradiance
(broadband sensors) tailored to a specific response, narrow band instruments such as filter radiometers with coarse spectral
resolution, and instruments performing high resolution spectral measurements – the most versatile but most challenging and
least robust instruments. Concerning the current UV monitoring measurement accuracy; The European reference UV
spectroradiometer (QASUME) is a traveling instrument which provides a common standard through inter-comparison on-site
(Gröbner et al., 2005; Hülsen et al., 2016). During the period 2000-2005 the QASUME visited 27 spectroradiometers sites.
Out of the 27 instruments, 13 showed deviations of less than 4% relative to the QASUME reference spectroradiometer in the
UVB (for 15 instruments in the UVA) for solar zenith angles below 75°. The expanded relative uncertainty (coverage factor
k=2) of solar UV irradiance measurements by QASUME, for SZA smaller than 75° and wavelengths longer than 310 nm, was
4.6% in 2002 – 2014 (Gröbner and Sperfeld, 2005), and has been  2 % since 2014 (Hülsen et al., 2016). For broadband
instruments, the current instrument uncertainties are summarized in (Hülsen et al., 2020; Hülsen et al., 2008). In 2017, 75
broad-band instruments measuring the UV index, the UVB or/and the UVA irradiance participated in the solar UV broadband
radiometer comparison in Davos Switzerland. Using the instrument/user calibration factors, the differences between the
datasets by the broad-band instruments and the reference (QASUME) dataset were within ±5 % for 32 (43 %) of the instrument
datasets, ±10 % for 48 (64 %), and exceeded ±10 for % 27 (35 %).
Although ground-based monitoring of solar UVR is more accurate than satellite retrievals, ground based stations are sparse,
and the only way for continuous monitoring of the UVR on a global scale is through satellites. In recent decades instruments
on-board satellites have provided the necessary data for estimates of UV irradiance reaching the Earth surface on a global scale





(Herman, 2010) and hence satellite-derived UVR climatological studies have been conducted (Vitt et al., 2020; Verdebout,
2004). The satellite UV irradiance record started with the Total Ozone Mapping Spectrometer (TOMS) on-board Nimbus-7 in
1978 and continued with Ozone Monitoring Instrument (OMI) on-board NASA's satellite EOS-Aura. The OMI retrieval
algorithm for surface UVR estimates was based on the experience gained from TOMS (Levelt et al., 2018; Levelt et al., 2006).
The early surface UVR retrieval algorithms from satellite data didn't account for the enhanced aerosol absorption in the UV
spectral range, resulting in overestimated values (Krotkov et al., 1998). A lot of scientific effort has been put into correcting
the products (Arola et al., 2009). TROPOspheric Monitoring Instrument (TROPOMI) onboard Sentinel – 5 Precursor (Lindfors
et al., 2018) is the current satellite instrument that provides the surface UVR product on a daily basis with global coverage,
including 36 UVR parameters. As the aforementioned instruments were installed onboard polar orbiting satellites, providing
global spatial coverage, the temporal resolution of the data is daily since there are only one or two overpasses per day for every
point. Geostationary satellites provide continuous (in time) measurements over wide areas. The geostationary meteorological
satellites Meteosat monitor the full Earth Disk including Europe and their frequent data acquisition of rapidly changing
parameters e.g. cloud is essential for estimating daily UV doses (e.g. Verdebout (2004)). Comparison of OMI surface UV
irradiance estimates with ground-based measurements for Thessaloniki, Greece showed that OMI irradiances overestimate
surface observations for UVB wavelengths by between ~1.5% to 13.5% in contrast to underestimated satellite values for UVA
wavelengths (Zempila et al., 2016). Results from the validation of TROPOMI surface UV radiation product showed that most
of the satellite data agreed within ± 10 % with ground-based measurements for snow-free surfaces (Lakkala et al., 2020).
Larger differences between satellite data and ground-based measurements were observed for sites with non-homogeneous
topography and non-homogeneous surface albedo conditions. The differences between GB and satellite UVR data are mostly
due to uncertainties in the input parameters to the satellite algorithm used to retrieve the UV irradiance at the surface. Based
on a recent study of Garane et al. (2019) a mean bias of 0-1.5% and a mean standard deviation of 2.5 – 4.5 % was found for
the relative difference between TROPOMI total ozone column (TOC) product and ground based quality assured Brewer and
Dobson TOC measurements.
In this study we present a UV-Index nowcast and forecast for the European region by using pre-calculated radiative transfer
model simulations in the form of analytical look-up tables (LUT) in conjunction with atmospheric input parameters from
satellites and high performance computing for instantaneous outputs of the order of 20 million grid points in less than 1 minute.
The Earth-Observation-based atmospheric inputs were retrieved operationally for the year 2017 and include the effects of
aerosols, clouds, solar and ground elevation, surface albedo and ozone. The aforementioned computing architecture formed a
novel system called UVIOS (i.e. UV-Index Operating System). The reliability of the UVIOS input and output parameters was
tested against ground-based measurements and an analytical sensitivity analysis was performed in order to quantify the
uncertainties and to provide information about the limitations and about the optimum operating conditions of the proposed
methodology.





In Section 2 we describe the UVIOS, the input data sources and the ground-based measurements used for the validation.
Section 3 presents the results in terms of model performance and factors that affect the UVIOS retrievals and the overall
accuracy. Finally, Section 4 summarizes the findings and the main conclusions of this study.
**2 Data and Methodology**
**2.1 The UV Index operating system (UVIOS)**
**2.1.1 UVIOS modelling**
The UVIOS system is a novel model that uses real-time and forecasted atmospheric inputs based on satellite retrievals and
modelling techniques and databases in order to nowcast and forecast the UVI with a spatial resolution of 5 km and a temporal
resolution of 15 minutes. The UVIOS calculation scheme is based on the libRadtran library of radiative transfer models (RTM)
(Mayer and Kylling, 2005) within which all the available inputs (i.e. solar elevation, cloud and aerosol optical properties,
ozone) can be integrated in real-time into the radiative transfer code and calculate the UVI for each pixel. Afterwards, post
processing correction for the elevation of each location and the surface albedo is also performed. In order to be able to simulate
the UVI for 1.5 million pixels in real-time we use pre-determined spectral solar irradiance LUTs based on the Libradtran RTM,
in combination with high performance computing (HPC) architectures that speed up the process of choosing and
interpolating/extrapolating the right combinations from the LUTs (Kosmopoulos et al., 2018; Taylor et al., 2016). The result
is the retrieval of UVI for 1.5 million pixels covering the European domain in less than 5 minutes after receiving all necessary
input parameters.
As mentioned the UVIOS architecture does not include a clear sky model and the subsequent calculation of individual sources
of UV attenuation, but instead it directly uses the following parameters: solar zenith angle (SZA), the aerosol optical depth
(AOD) and other aerosol optical properties (e.g. single scattering albedo (SSA) and Ångström exponent (AE)), the total ozone
column (TOC), the cloud optical thickness (COT), as well as the surface elevation (ELE) and the surface albedo (ALB) as
RTM inputs. Table 1 presents the EO data used as inputs for the UVI real time simulations, their description and sources. The
Meteosat Second Generation (MSG4) cloud microphysics includes the nowcasted cloud optical thickness (COT) at 550 nm,
and cloud phase (CPH) obtained at a spatial and temporal resolutions of 5 km (average, depending on latitude) and 15 minutes,
respectively. Typical values of other cloud properties (e.g. cloud height, cloud thickness) have been assumed based on the
cloud type (information which is also available from MSG) (for more detailed information see Taylor et al. (2016). The 1-day
forecast CAMS aerosol optical depth (AOD) at 550 nm is obtained at a spatial and temporal resolutions of 40 km and 3 h,
respectively and the monthly aerosol optical properties obtained from Aerocom (Kinne, 2019) includes single scattering albedo
(SSA) and Ångström exponent (AE) at 1° x 1° (latitude x longitude) spatial resolution. Solar elevation is taken from the
Astronomical model (NREL) (5 km – 15 minutes) and climatological surface albedo (ALB) is retrieved from Copernicus



Global Land Service (CGLS) (1 km – 12 days). Surface elevation (ELE) is obtained from the Digital Elevation Model (DEM)
of NOAA. The Tropospheric Emission Monitoring Internet Service (TEMIS) 1-day forecast of total ozone column (TOC) is
at a spatial resolution of 1º x 1º – 1 day with assimilated ozone fields from GOME-2 (METOP-B). We have to mention also
here that the selection of the RTM inputs has been decided based on their real-time availability.

**2.1.2 UVIOS real-time processing concept**

The LUT approach, despite its large size (almost 2.5 million spectral RTM simulations for clear and all sky conditions)
(Kosmopoulos et al., 2018), still provides estimates at discrete input parameters values. To overcome this mathematical issue,
we performed a multi-parametric interpolation technique to correct the input-output parameter intervals. This solution is
computationally more costly than a continuous function-approximation model, i.e. a Neural Network (NN) model
(Kosmopoulos et al., 2018), but the accuracy improvement is significant. Indicatively, using a test set of 1 million RTM
simulations for UVI from the developed LUT, we applied the NN developed in Kosmopoulos et al. (2018) and found a mean
execution time of around 144 seconds followed by a mean absolute error (MAE) of 0.0321, while by using the proposed
UVIOS multi-parametric interpolation exploiting the HPC and distributed computing benefits we found for the same test set
an execution time of 295 seconds with a MAE of 0.0001. The inclusion of many parameters (in this study we incorporated
eight, i.e. AOD, SZA, TOC, COT, ELE, ALB, AE, SSA) with small step sizes dramatically increase the LUT size, followed
by high computing requirements for the multi-parametric interpolation/extrapolation procedures.
For the UVIOS simulations performed in this study, a 32-core UNIX server was used equipped with 256 Gb of RAM and 12
Tb of storage system working in a RAID10 architecture. The combination of the HPC with the analytical LUTs, which were
developed by using the libRadtran RTM, allow a high speed multi-parametric interpolation and polynomial reconstruction
(Gal, 1986) to increase accuracy between the LUT records following a mathematical equation relating the UVIOS outputs to
the EO inputs.
An example of the UVIOS input output data is presented in Figure 1 through a flowchart illustration of the modelling technique
scheme. The inputs, including the solar and surface elevation, albedo, aerosol, ozone forecasts and the cloud observations as
described in Table 1, are fed to the real-time solver that results in spectrally weighted output of UVI for the European region.
Figure 2 shows the memory usage and error statistics for a range of different LUT sizes. The LUT error decreases as the LUT
size increases, regardless of the function being approximated. The LUT sizes in Figure 2 fit into cache on our HPC
environment, thus performance in terms of processing speed and overall output accuracy vary only slightly between the table
sizes shown. In our case, UVIOS shows that LUT transformation can provide a significant performance increase without
incurring an unreasonable amount of error, provided there is sufficient memory available. We note that the cache size is a
critical factor for LUT performance, while under a HPC environment practically there is no limit. Such techniques can be
implemented in hardware with distributed computing that operates in parallel to provide optimum performance.



## 2.2 Earth-Observations

The Cloud Optical Thickness (COT) data from Meteosat was used, whose retrieval algorithm is based on 0.6 and 1.6 micron channel radiances of Meteosat SEVIRI. MSG products have been described in Derrien and Le Gléau (2005) and the MétéoFrance (2013) technical report. The COT impact uncertainty on UVI deals with the MSG COT reliability and accuracy and hence introduces errors into the UVIOS simulations (Derrien and Le Gléau, 2005; Pfeifroth et al., 2016). In addition, comparison principles of (point) station UVI measurements with a 5 km MSG COT matrix are possibly responsible for at least part of the observed deviations (e.g. Kazadzis et al. (2009a)). For instance, when a MSG pixel is partly cloudy, the ground measurements of UVI could fluctuate more than 100%, depending on whether the sun is visible or whether clouds attenuate the direct component of the solar irradiance. The result is that in cases of partly covered MSG pixels and in the absence of clouds between the ground measurement and the sun, the ground truth UVI would be much higher than the UVIOS one. Of course, the presence of small clouds which have not been identified by MSG and cover (part of) the sun disk, is plausible as well, consequently causing an overestimation of the modelled UVI (Koren et al., 2007). Furthermore, sensors onboard geostationary satellites suffer from the parallax error, which contributes to the spatial errors of the images and the overall uncertainty of the products (Bieliński, 2020; Henken et al., 2011). The error depends on the altitude of the cloud and the viewing angle (parallax errors are more significant for high viewing angles).

UVIOS calculations at high solar zenith angles (>75 deg) are retrieved assuming cloudless skies since the MSG COT product is not available in these conditions. This has an effect on the quality of the UVIOS overall performance at high solar zenith angles. However, since such measurements are associated with very low UV Index (<1), this inconsistency does not affect UVIOS UV Index results associated with dangerous effects on human health. There is more discussion in the next section on how we use these data for the UVIOS validation.

For the total aerosol optical depth, we used 1-day forecast data from the Copernicus Atmospheric Monitoring Service (CAMS) as the basic input parameter. These forecasts are based on the Monitoring Atmospheric Composition and Climate (MACC) analysis and provide accurate data of aerosol optical depth (AOD) at 550 nm with a time step of 1 h and spatial resolution of 0.4°.

For aerosol single scattering albedo properties climatological values from MACv2 aerosol climatology (Kinne, 2019) was utilized. Monthly means of single scattering albedo at 310nm were acquired from global gridded data at a 1° x 1° spatial resolution. Also, in order to derive the Angstrom exponent, monthly means of AOD at 340nm and 550nm were used. The calculated Ångström exponent was then applied to the 550 nm AOD (from CAMS) in order to get AOD in the UV.

The surface albedo data were obtained from the Copernicus global land service (CGLS: Geiger et al., 2008; Carrer et al., 2010). As a global surface ALB product is not available in the UV region, for this study we have used the climatological product of CGLS (in the visible range) as follows: based on the findings of Feister and Grewe, 1995; we used a UV albedo of 0.05 for non-snow cases and a UV ALB equal with CGLS when CGLS exceeded 0.5 (snow cover). The total ozone column forecasts were obtained from Tropospheric Emission Monitoring Internet Service (TEMIS) which is a near- real time service which uses



the satellite observations of total ozone column by the Global Ozone Monitoring Experiment (GOME) and SCIAMACHY
assimilated in a transport model, driven by the European Centre for Medium-Range Weather Forecasts (ECMWF) forecast
meteorological fields (Eskes et al., 2003). The elevation data was obtained from the 5-minute Gridded Global Relief Data
(ETOPO5) database, which provides land and seafloor elevation information at a 5-minute latitude/longitude grid, with a 1-
meter precision in the region of Europe and is freely available from NOAA (NOAA, 1988). Figure 3 shows an example of the
input-output UVIOS parameters. An extensive validation of the MACC analysis and forecasting system products were
performed by Eskes et al. (2015). The aerosol optical properties were validated against 3-year (Apr. 2011 – Aug. 2014) near
real time level 1.5 Aeronet measurements and for AOD at 550 nm an overall overestimation was exhibited. Due to dedicated
validation activity of the MACC service a validation report that covers the time period of this study (Eskes et al., 2018) is also
available, presenting an overall positive bias during 2017. This overestimation of AOD at 550 nm may explain some of the
UVI underestimation under clear sky conditions (see section 3.2.2).
**2.3 Ground-based measurements**
In order to validate the UVIOS results 17 ground based stations were selected, for which measurements of the UVI were
available during 2017. The stations are shown in Fig. 4. Comparisons were performed with a 15-minute step. The ground based
measurements were obtained from spectrophotometers (Brewer), spectroradiometers (Bentham), filter radiometers (GUV) and
broadband instruments (SL501 and YES) as Table 2 shows. I Note that UV data in table 2 has been calibrated, processed and
provided directly by the responsible scientists for each station. References wherein more information for the data quality of
particular instruments can be found are also provided. Brewer spectrophotometers measure the global spectral UV irradiance
with a step of 0.5 nm, and a resolution which is approximately 0.5 nm (usually between 0.4 and 0.6 nm). Depending on their
type the spectral range is usually 290-325 nm (MKII, MKIV) or 290-363 nm (MKIII,). Since Brewer spectrophotometers
measure the spectrum up to a wavelength which is shorter than 400 nm, extension of the spectrum up to 400 nm in order to
calculate the UV index is usually achieved using empirical methods (e.g. (Fioletov et al., 2003; Slaper et al., 1995)). The
additional uncertainty in the UVI due to the latter approximation is well below the overall uncertainty in the measurements.
Bentham spectroradiometers measure the whole UV spectrum (290 – 400 nm) with step and resolution which can be
determined by the operator. The spectra from AOS and LIN (measured by Bentham spectroradiometers) used in this study
have been recorded with a step of either 0.25 or 0.5 nm and a resolution of ~ 0.5 nm.  The Brewer Spectrophotometer measures
the total column of ozone using the differential absorption method, i.e. measuring the direct solar irradiance at four wavelengths
and then comparing the intensity at wavelengths that are weakly and strongly absorbed by ozone (Kerr et al., 1985). Brewer
TOC measurements are used in the present document to validate the TEMIS forecasts. The Ground-based Ultraviolet (GUV)
instrument is a multichannel radiometer that measures UV radiation in five spectral bands having central wavelengths as 305,
313, 320, 340 and 380 nm. However, in addition to UV irradiances, other data that can be obtained from GUV instruments are
total ozone and the cloud optical depth (Dahlback, 1996; Lakkala et al., 2018). GUV measurements are used for LAN station





of Norway. At stations AKR, INN and VIE, the surface UV was measured using Solar Light (SL) 501 radiometers. It provides
direct observation of UV index with a frequency of one minute. The Yankee Environmental System (YES) has been used for
VAL station. The low latitude stations include AKR, ARE, ATH, ROM, THE, and VAL. AKR has minimum altitude of 23 m
and VAL has maximum altitude of 705 m above sea level. The middle latitude locations are AOS, DAV, INN, BEL, LIN,
MAN, UCC, and VIE among which the minimum altitude is 10 m in LAN and maximum altitude is in DAV at 1610 m above
mean sea level. HEL, LAN, and SOD represent the high latitude zone, with HEL having an altitude of 48 m and SOD an
altitude of 185 m above mean sea level (Table 2).
A summary of basic climatic information for the validation locations was obtained from the Köppen climate classification
(Chen and Chen, 2013) and it is summarized here. THE, AKR, ARE, ROM, ATH and VAL have a Mediterranean climate
comprising of mild, wet winters and dry summers. MAN experiences maritime climate (cool summer and cool, but not very
cold, winter). AOS, UCC, LAN, BEL, HEL, LIN and VIE experience humid continental climate with warm to hot summers,
cold winters and precipitation distributed throughout the year. DAV and INN experience boreal climate characterised by long,
usually very cold winters, and short, cool to mild summers. SOD has subarctic climate having very cold winters and mild
summers.
**2.4 Evaluation methodology**
The time series period covers the whole year 2017 at 15-min intervals, following the MSG available time steps. A
synchronization between the UVIOS simulations and the ground-based measurements was performed in order to match the
15-min intervals of UVIOS to the measured data. The UVIOS data availability is 93%, while for the ground stations it reaches
almost 79% enabling a direct UVI data comparison of 77% of the 2017 time steps. For the comparison we used the closest
instrument measurements to the 15-min intervals with a maximum deviation of 3 minutes in order to avoid solar elevation and
cloud presence mismatches. Additionally, the UVIOS comparisons included measurements up to 70 degrees SZA. The
rationale for this cutoff was that UVIOS retrievals at high SZA are retrieved as cloudless as COT is unavailable from MSG.
In addition, the comparison is also impacted by limitation of the horizon of ground-based (GB) sites (e.g. Davos, Innsbruck,
Aosta) where the diffuse component and in some cases the direct component of solar UV irradiance are affected by obstacles
(mountains) on the horizon. The contribution of this mainly diffuse irradiance to the total budget is a function of solar elevation
and azimuth (day of the year) and also cloudiness.  This explains some of the deviations in the results as the UVIOS retrieves
UVI assuming a flat horizon. Clear sky conditions where defined as the UVIOS retrieval where MSG COT equals to zero.
Further discussion on the uncertainties introduced by this choice is mentioned in the cloud effect section.
Most of the comparisons have been performed using the absolute (mean bias or median) UVI differences (model –
measurements). In addition, median values of the percentage differences (100* (model – measurements)/measurements) have
been used. UVIOS estimations were also evaluated in terms of mean bias and root mean square error (MBE and RMSE,
respectively), defined as follows:





$$\text{MBE} = \bar{\varepsilon} = \frac{1}{N}\sum_{i=1}^{N}\varepsilon_i \qquad (1) \qquad \text{RMSE} = \sqrt{\frac{1}{N}\sum_{i=1}^{N}\varepsilon_i^2} \qquad (2)$$

Where $\varepsilon_i = x_f - x_o$ are the residuals (UVIOS errors), calculated as the difference between the simulated values ($x_f$) and the
ground-based values ($x_o$), and where N is the total number of values. MBE quantifies the overall bias and detects whether the
UVIOS overestimates (MBE>0) or underestimates (MBE<0). RMSE quantifies the spread of the error distribution. Finally,
the correlation coefficient (r), as well as the coefficient of determination ($R^2$) were used to represent the proportion of the
variability between modeled and measured values.

## 3 Results

### 3.1 Performance of the UVIOS technique; result overview

Fig. 5 presents a Taylor diagram with the overall UVIOS accuracy for all ground stations under all sky and clear sky conditions
as a function of the correlation coefficient, normalized standard deviation and RMSE. Under clear sky conditions, the UVIOS
shows the best results with a correlation coefficient (r) close to 0.98 for THE, ROM, AKR, ARE, AOS, and VAL, while the
RMSE is below 0.5 UVI for the majority of stations. Under all sky conditions, the error increases with higher uncertainties at
MAN, DAV and SOD, which is probably caused by misclassification of cloudy pixels (see also the Appendix A section). For
both, clear sky and all sky conditions, the RMSE of the absolute UVI differences between the model and the measurement is
close to 0 (within ± 0.5 UVI) for all stations. Most of the comparisons are provided comparing UV Index median values and
not percentages. Averaging percentage differences does not provide a fair estimation of the model as such differences in most
of the cases are not represented by a normal distribution and also small UVI differences (especially in high cloudiness instants
and high solar angles -  e.g. differences less than 1 UVI having UVI values less than 1) can differ by up to 500%. Thus, we
have focused on absolute differences in order to have a more representative assessment of the actual effect (UV Index) and its
results based on low (less than 0.5), moderate (0.5-1) and high (more than 2) UVI differences between UVIOS and the ground
based measurements. However, in the Appendix A we also provide such differences in percent.
In Table 3, U1.0 and U0.5 represent the percentage of cases with absolute differences between modelled and ground based
UVI measurements within 1 and 0.5, respectively, for all comparisons between the 15-minute model retrievals and the
corresponding ground-based measurements.  As shown in Table 3, for all stations and for both, clear- and all sky conditions,
differences were within 0.5 UVI for at least 70% of the cases. Under clear sky conditions, AOS, BEL, HEL, LAN, LIN, SOD
and THE had above 90% of U0.5 cases, while others have 75-90% of U0.5 cases. All stations but DAV had above 90% of
U1.0 cases for clear skies, while the correlation coefficients for most of the stations were above 0.9 (exceptions are ATH and
MAN). For all-skies differences were within 1 UVI for 90% of the cases for all stations with the correlation coefficients
exceeding 0.9 for most of them (exceptions are DAV, MAN and SOD). Median differences for all skies for every station were
well within $\pm 0.2$ UVI, with the 25-75 percentiles being within $\pm 0.5$ UVI and the 5-95 percentiles within $\pm 1$ UVI. . For clear





skies the corresponding values are $\pm 0.1$, $\pm 0.4$ and $\pm 0.8$ respectively. In the following sections we try to investigate the factors
that contribute to the differences between UVIOS and ground-based measurements.
**3.2 Factors affecting UVIOS retrievals**
**3.2.1 Ozone effect**
All the available collocated Total Ozone Column (TOC) measurements for the stations used in UVIOS evaluation have been
obtained from the WOUDC (https://woudc.org/) database. In this database 8 out of 17 UVIOS evaluation stations (AOS, ATH,
DAV, MAN, ROM, SOD, THE and UCC) were found, providing TOC GB measurements. TOC comparison has been
performed by calculating daily means of GB measurements and the TOC from TEMIS. In order to quantify the effect of the
uncertainty of the forecasted TOC used as input at UVIOS we have calculated the mean differences of the forecasted and
measured TOCs and used a radiative transfer model to investigate their effect on UVIOS retrieved UVI. Table 4 shows the
mean differences in D.U. from TEMIS TOC (used as inputs in UVIOS) as compared to the WOUDC ground-based
measurements for one year of comparison data. It is seen that for the stations AOS, DAV, MAN and UCC the values of the
TEMIS observations are higher as compared to the ground-based measurements while for the other stations TEMIS
observations are lower. The negative bias is seen to be highest for ROM station (-9.9) and the positive bias is highest for AOS
station (7.6). Part of the large differences over the complex terrain sites can be explained by the difference between the actual
altitude of the station and the average altitude of the corresponding grid points of TEMIS. For example, for AOS the average
altitude of the pixel is 2000 m while the real altitude of the station is 570 m, resulting in an underestimation of the tropospheric
column of ozone by TEMIS. In general, differences can be explained by the combined effects of uncertainties in TOC retrieval
from satellite and GB platforms (Rimmer et al., 2018; Boynard et al., 2018; Garane et al., 2018). Figure 6 shows the effect of
this TOC bias on the calculated UVIOS. As seen in Table 4, there is a mix of small underestimation and overestimation cases
in the TOCs used within UVIOS, with average absolute differences of 4-5 DU. Worst TOC UVIOS inputs were found in AOS
and ROM (7.6 and -9.9 DU) leading to maximum (at 30 degrees SZA) differences in UVI of -0.22 and 0.3 for AOS and ROM,
respectively. In general, in most of the cases UVI mean differences are less than 0.1. It has to be noted that the TOC differences
have a larger impact when expressed in percent at higher SZAs, while in Figure 6 higher absolute differences for low SZA's
are associated with higher UVIs at these SZAs. Detailed comparisons for each station are shown in the Appendix A figures.
**3.2.2 Aerosol effect**
Aerosol optical depth measurements used for the UVIOS aerosol input evaluation have been collected from the AERONET-
NASA web site (Giles et al., 2019) for 12 out of our total 17 stations (AKR, ARE, ATH, DAV, HEL, LIN, ROM, SOD, THE,
UCC, VAL and VIE. AERONET (level 2, version 3) values of AOD at 500 nm were interpolated at 550nm using the
AERONET derived 440-870nm Angstrom exponent for each individual measurement. In order to compare those


measurements with CAMS forecasted AOD used for the UVIOS their daily means were derived. The comparison of forecasted
and measured daily means was based on all available data due to gaps in the AERONET time series. The AOD MBE and
RMSE statistical scores are shown in Table 5 in absolute units and correlation coefficient as well. All the stations have a mean
positive bias up to 0.071 except UCC which is showing a mean negative bias of 0.007. The comparison of all individual
stations with CAMS data used as inputs on UVIOS showed that under all cases CAMS AOD is higher than that from
AERONET with a mean difference of 0.07 at 550nm. The correlation between the modeled and the measured values varies
from 0.10 for VIE to 0.91 for ARE with most of the stations showing the correlation coefficient above 0.7. As in the case of
the TOC, AOD CAMS data are forecasts from the previous day and real time WOUDC or AERONET level 2.0 data do not
exist. Although real time TOC (and in due course AOD in the UV) is available from Eubrewnet (López-Solano et al., 2018;
Rimmer et al., 2018), it is only for particular locations and not for the whole European domain. Thus, the only choice in
providing for a real time UV Index for Europe is using the CAMS (for AOD) and the TEMIS (for TOC) data.
In order to evaluate the effect of AOD on UVI, UVI differences between the UVIOS using both AOD datasets (CAMS and
AERONET) as UVIOS inputs were analyzed. Figure 7 shows the mean bias error of the CAMS – AERONET AOD impact on
UVI for all stations with available ground based AOD data as a function of SZA together with the uncertainty range (± 1 σ).
It can be seen that UVIOS with CAMS AOD input underestimates UVI compared to the UVIOS with AERONET data, except
for the UCC station. This is consistent with CAMS overestimations of AOD compared to the AERONET measurements,
except for the station UCC as shown in Table 5. Higher aerosol levels in the atmosphere tend to lower the UVI. Highest
difference in UVI is observed for the stations HEL, SOD, VIE. Since, the aerosol level at the stations HEL and SOD is very
low, the percent difference between the AOD from CAMS and AERONET is larger for these stations (although the absolute
difference is similar) relative to stations with higher AOD, leading to higher differences in the UVI. Aerosol content for VIE
is higher than HEL and SOD but still within 0.2 which might be the reason for the higher UVI difference. In terms of SZA, it
is observed that the mean bias decreases with an increase in the SZA as the values of UVI also decrease with SZA and the
most deviation is for station VIE which is consistent with the poor correlation between the CAMS forecasted input and the
measurements for this station as seen from Table 5.
The use of single scattering albedo in the UV region is a difficult task and many studies have shown that such measurements
need extra effort and it is not possible to perform them worldwide (Arola et al., 2009; Kazadzis et al., 2016; Raptis et al.,
2018). The monthly values of the single scattering albedo used in UVIOS for the UV region were derived from the MACv2
database at the 310 nm wavelength (Kinne, 2019). Fig 8 shows the intra annual variability of SSA for the 17 stations. For all
stations, SSA values range from 0.76 to 0.93, with most of them having SSA values between 0.83 to 0.93, and relatively small
variability. In contrast, there are stations like ARE, BEL, INN, LIN, VIE and THE which have relatively smaller SSA values
(0.76-0.9) and greater variability than the other stations.




### 3.2.3 Albedo effect


Surface albedo at UV wavelengths is small (2 – 5%) for most types of surfaces (Feister and Grewe, 1995; Madronich, 1993)
except for features like sand (with a typical albedo of ~0.3) and snow (up to 1 for fresh snow) (Meinander et al., 2013; Myhre
and Myhre, 2003; Vanicek et al., 2000; Henderson-Sellers and Wilson, 1983). Renaud et al. (2000), found an enhancement of
about 15 to 25% in UVI for clear-skies and snow conditions due to the multiple ground-atmosphere reflections and this relative
increment was about 80% larger for overcast conditions. The combined effect of aerosols and snow lead to an enhancement of
about 50% in UVI in cloud-free condition for moderately polluted atmospheres (Badosa and Van Weele, 2002). Fig. 9 presents
the effect of surface albedo on the UVI percentage difference (i.e. for various albedo values under clear sky conditions) as a
function of SZA. It is observed that the UVI percentage difference increases almost linearly with albedo for a particular SZA
and the variation is found to be almost identical for all SZA. This indicates that the UVI percentage difference is independent
of the SZA and increases with surface albedo.
Uncertainties introduced in UVIOS from the use of a constant surface albedo value of 0.05 for non-snow conditions are quite
low. For the case of albedo values used for snow conditions based on the CGLS monthly mean product uncertainties can be
related to: the small difference of UV and visible albedo values; the fact that the CGLS provides an albedo of a certain area
around the station that does not necessarily coincide with the "effective" albedo area affecting UV measurements; and finally
that the monthly albedo product represents a monthly average while a real time CGLS product represents the last 12 days
(dynamically changing albedo). In order to investigate this last point we have compared the UV effects from the use of the two
albedo datasets for DAV station. In Fig. 10, the effect of surface albedo correction is shown for the Davos station, for a period
with snow cover and low percentage cloudiness. The climatological and the dynamically changing albedo are presented in
terms of percentage differences between modelled and ground measurements as a function of SZA. In the case of climatological
albedo, most of the percentage difference between forecasted and the measured UVI value is found to vary from -30% to 10%
for SZA between 20° to 70°, showing more underestimation than overestimation from the UVIOS simulations. Similarly, in
the case of dynamically changing albedo, most of the percentage difference between forecasted and the measured UVI value
is found to vary from -20% to 10% for SZA between 20° to 70°. The mean percentage difference between the results using the
two different albedo inputs is -2.76% in terms of accuracy improvement. However, beyond 70 degree SZA, there is a huge
variation in the percentage difference with mostly underestimations from the UVIOS simulations (not shown in Fig. 10).

### 3.2.4 Cloud effect


For the evaluation we used measurements at SZA lower than 70 degrees, based on the lack of cloud input from MSG for higher
SZAs. The lack of MSG data results in an overestimation of UVIOS in high SZAs and the UVI is systematically overestimated
for long periods during winter at high latitude regions when SZA does not get below 70 degrees during the day. However, this
overestimation is low in terms of absolute UVI and does not usually exceed 0.2 UVI because maximum UVIs at such SZAs
rarely exceed UVI=1.





COT retrieved from the MSG satellite has been used as input for the UVIOS together with typical optical properties of the
clouds as discussed in Sect. 2.1. The evaluation of all stations for cloudless and cloudy conditions can be seen in Figure 11
that shows the relative frequency distribution of all stations (colours) and the mean (black line) for cloudless (upper plot) and
cloudy conditions (lower plot). Mean bias error of the modeled by UVIOS and measured UVI for all- and clear sky conditions
and the percentage of clear sky time steps data is presented in Figure 12. The mean bias for clear sky conditions is found to be
less than that for the all sky condition for the stations AKR, ATH and THE (having most days of the year being cloudless as
the clear sky percentage is above 70%). The MBE for DAV, LIN and MAN is less for clear sky relative to all sky conditions
even though most days of the year are cloudy (clear sky annual percentage less than 45%) at the particular stations. While,
stations BEL, HEL, INN, LAN, SOD, UCC and VIE, that have mostly cloudy skies throughout the year (clear sky annual
percentage less than 50%), are having more MBE for clear sky conditions than the all sky condition. This can be due to the
erroneous classification of a cloudy sky as clear sky, which is also discussed in the following section. MBE is also larger for
AOS and ARE which have mostly clear skies throughout the year. Stations ROM and VAL have comparatively much smaller
MBE for clear sky conditions.
As shown in Table 6 there are 45.4% of cases with underestimations and 54.6% cases with overestimations for cloudless
conditions (COT=0). For all the other cases, overestimations (62.5%) are more predominant than underestimations (37.5%).
The difference in the modelled and the measured values goes beyond $\pm 1$ UVI for only 5.1% cases for cloudless condition and
14.7% for all other cases. In general, under cloudy conditions, UVIOS shows an overestimation for UVI in contrast to the
ground measurements. One explanation for the overestimations could be the erroneous determination of COT from MSG above
the ground-based stations, giving cloud input that can be overestimated or underestimated. The results show that there is a
general tendency for a small underestimation of MSG COT that leads to a systematic but small UVIOS UVI overestimation
under cloudy conditions. Another possible explanation is the spatial representativeness of MSG COT. The MSG COT
determination is available at 5 by 5 km pixels that may differ from the actual situation of the cloud prevailing above the station,
especially in broken cloud conditions and a case when it blocks the direct radiation from the sun. Moreover, for lower solar
elevations, the direct sun irradiance can be blocked by cloud in neighbouring pixels. The first effect has been explored in the
relative frequency distribution of Figure 11 that shows a higher number (~ 63%) of data on the right of the zero UVI difference
vertical line for cloudy skies. When comparing data outside the 0.5 and 1 difference limits we also see that $1 - 4$ times more
data show a UVIOS overestimation as compared to the clear sky case. This shows that in general there is a small (in UVI
terms) but significant UVIOS overestimation for non-zero COT conditions. Moreover, for clear skies, as determined from the
MSG, we observe a less pronounced UVIOS overestimation that corresponds to the fact that even if MSG defines the situation
as completely cloudless, in reality there may be some cases where clouds near the GB station affect the measured UVI. This
effect is easier to understand when showing these differences as a function of solar zenith angle which is explored through
Figure 13. It is observed that the absolute difference between the modelled and the measured values decreases with increasing
solar zenith angle and most of the difference lies within $\pm 4$ UVI. The seasonal variation of the percentage UVI difference as



a function of SZA shows that while absolute UVI is small in winter the percentage difference is higher compared to other
seasons.
Figure 14 (a) shows the surface horizontal distance as a function of the cloud height and the SZA required for the cloud to
block the sun. 14 (b) shows the scatter of the UVI difference under clear sky conditions for all stations as a function of SZA.
It is observed that there is an obvious pattern of scattered data for UVI differences higher than 1.5 compared with the ones for
differences less than -1.5. These data represent UVIOS overestimation for UVI retrievals due to the underestimation of the
cloudiness just above the stations. These data illustrate the well-known spatial representativeness issues whereby a COT value
for a satellite grid is not fully representative of a point measurement station. In addition, absolute and percentage relative
differences are shown in Fig. 14 (c) and (d) respectively for SZA up to 65 degrees. The differences between the UVIOS and
the GB UVI decreases in absolute level but increases in percent with an increase in SZA. This is due to the decrease of UVI
with increasing SZA. Modelled and the measured UVI difference is close to zero both for mean and median values. For SZA
below 30 degrees differences are 0 to -0.2, while 20 to 80 percentiles range from -0.6 to -0.2. Percentage difference increases
with SZA as absolute UVI decreases with the 20 to 80 percentiles showing differences between -10% and 10%
**3.2.5 Surface elevation correction**
Fig. 15 presents the effect of surface elevation on UVI as a function of the percentage difference for various total ozone
columns. The UVI percentage difference is found to increase almost linearly with the increase in elevation for a particular total
ozone column. The percentage difference is similar for all ozone columns up to 1km, after which the differences with ozone
column become more apparent. That is, at a particular elevation, the percentage difference is higher for less total ozone column.
A 1% fluctuation (decline or increase) in column ozone can lead to about a 1.2% fluctuation (increase or decline) in the UV
Index (Fioletov et al., 2003; Probst et al., 2012). Indicatively, the surface elevation correction in UVI for the DAV station (due
to UVIOS input deviation from to actual elevation) was of the order of 15%, while for INN and AOS it was 6% and for the
VAL station close to 8%.
**4. Summary and conclusions**
In this study, a fast RTM model of UVI, the so-called UVIOS, using inputs of the SZA, aerosol optical depth, total ozone
column, cloud optical depth, elevation and surface albedo that implicitly includes temporal effects and the effect of cloud and
aerosol physics, allows for the generation of high-resolution maps of UVI. Ground based measurements of UV are the most
accurate way to determine this important health related parameter. However, such stations are sparse and hence, satellite
observations can be used in order to have a nowcasted UV service. To date polar orbiting satellites like TOMS, OMI and
recently TROPOMI provided a global UV dataset with a major disadvantage being the temporal resolution (one measurement
per day). This, combined with the large temporal variability of clouds can lead to huge deviations from reality when a single





daily measurement is included. Geostationary satellite, MSG, have been used in order to try to improve on such limitations
using cloud information every 15 minutes.
Comparison of the forecasted and the ground-based measurements indicated that at least 70% and 80% of comparisons were
within 0.5 UVI difference for all sky condition and clear sky, respectively. The mean differences TEMIS TOC and the ground
measured TOC from the WOUDC for one year of comparison data showed that TEMIS tends to slightly overestimate the TOC
for some stations along with underestimating it for other stations. While, in general, in most of the cases UVI mean differences
are less than 0.1, the TOC differences have a larger impact in percent UVI differences at higher SZAs. Such small differences
can also be the result of daily TOC variation not captured in TEMIS.
CAMS AOD seems to be slightly overestimated as compared with AERONET data that leads to a UVIOS underestimation.
CAMS data are found to overestimate the AOD from AERONET measurements with a mean difference of 0.07 at 500 nm.
All the stations have a mean positive bias up to 0.071 except one station that had a mean negative bias of 0.007. The analysis
of the impact of the mean bias error of the CAMS – AERONET AOD impact on UVI for all stations showed that the mean
bias decreases with an increase in the SZA as the values of UVI also decreases with SZA. The greatest deviation is for station
VIE which is consistent with the poor correlation between the CAMS forecasted input and the measurements for this station.
The real time data provision approach of UVIOS requires using a maximum of one-day ozone and aerosol forecast using the
TEMIS and CAMS service respectively.
Cloudy conditions show high percentage differences but low UVI differences, and have a general tendency to lead to a UVIOS
overestimation. It was found that 45.4% of cases have underestimations while 54.6% cases have overestimations for the
cloudless conditions, while overestimations (62.5%) were more predominant than underestimations (37.5%) for all the other
cases. In general, UVIOS showed an overestimation for UVI in contrast to the ground measurements under cloudy conditions
with the difference in the modeled and the measured values going beyond ±1 for 5.1% cases for cloudless conditions and
14.7% for all other cases. At individual stations the results for cloudless sky conditions, which are the most important for
health related issues, showed good agreement. In general, ~85% of all and 95% of cloudless cases are within 1 UVI difference.
The relative percentage biases can be large for low UVI cases due to clouds or at high SZAs, above 75°, due to the absence of
accurate information for clouds. The results show that there is a general tendency of small underestimation of MSG COT that
leads to a systematic but small UVIOS overestimation under cloudy conditions. Another possible explanation is the spatial
representativeness issues between a satellite and a single point on the ground.
SSA validation is difficult to perform as there are no systematic SSA measurements in the UV region. Using climatological
surface albedo has little impact at low albedo sites but mainly leads to underestimations in UVIOS simulations for high albedo
situations (snow cover). Most of the percentage difference between forecasted and the measured UVI values varied from -30%
to 10% for SZA between 20° to 70° (climate albedo), while it was found to vary from -20% to 10% for dynamically changing
albedo. Since high surface albedo conditions correspond to winter months (i.e. high SZAs and relatively low UVI) for the
stations used in the study, the corresponding absolute differences in the UVI are generally smaller than 2 UVI. However, there
was a huge variation in the percentage difference beyond 70 degree SZA with mostly underestimations from the UVIOS



simulations. Finally, for uncertainties in elevation inputs, the UVI percentage difference is found to increase almost linearly
with the increase in elevation for a particular total ozone column and beyond that, it is seen that the rate of increase in the
percentage difference decreases with increase in the total ozone column.
UVIOS system forms a novel tool for widespread estimations of UVI using real-time and forecasted EO inputs. UVIOS utilizes
the MSG domain with high spatiotemporal resolution, producing outputs within acceptable limits of accuracy for UV health
related applications. It captures basic cloud features and all major atmospheric and geospatial parameters that affect UVI.
Under cloudless conditions it performs to within the uncertainty of the ground based measurements to which it has been
compared. Further development and improvement of the model can be achieved in the future. Meteosat Third Generation
satellites are expected to be launched in the following years and give aerosol and cloud products which would improve the
performance of nowcast and forecast UV models when used as inputs.
**Author contribution**
PGK was responsible for the design of the study and the whole analysis, with support from SK, AWS, PIR, KP, IF and AM.
PGK and SK are the developers of UVIOS. All authors contributed to editing the paper.
**Code/Data availability**
All data used as inputs to the UVIOS system are open access, while all data sets produced by the UVIOS for the purposes of
this paper can be requested from the corresponding author. The ground-based measurements can be requested from the PIs of
the stations. The UVIOS suite of algorithms and LUTs can be used for various applications after consultation with the
corresponding author.
**Competing interests**
The authors declare that they have no conflict of interest.



## Acknowledgements

We acknowledge the Eumetsat SAFNWC, the Copernicus and TEMIS services as well as the Aerocom and GOME teams for providing all the necessary data used in this study. We would like to thank the 17 site instrument operators and technical staff that made the ground based measurements feasible.

## Financial support

This research has been partly funded by the European Commission project EuroGEO e-shape (grant agreement No 820852).

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












**Table 1:** UVIOS model input parameters


| Parameter | Description<br>(spatial – temporal resolution) | Source | Reference |
|---|---|---|---|
| Cloud microphysics | Nowcast cloud optical thickness (COT), cloud phase (CPH) (5 km – 15 minutes) | Meteosat Second Generation (MSG4) NOA Antenna | (MétéoFrance, 2013) |
| Aerosol optical depth | 1-day forecast aerosol optical depth (AOD) (40 km – 3 hours) | Copernicus Atmosphere Monitoring Service (CAMS) – FTP access | (Eskes et al., 2015) |
| Aerosol optical properties | Single scattering albedo (SSA), Angstrom exponent (AE) (1 x 1 degrees – 1 month) | Aerosol Comparisons between Observations and Models (Aerocom) | (Kinne, 2019) |
| Solar elevation | Solar zenith angle (SZA) (5 km – 15 minutes) | Astronomical model In-house software (NOA) | (Reda and Andreas, 2008)) |
| Surface albedo | Surface albedo (ALB) (1 km – 12 days) | Copernicus Global Land Service (CGLS) | (Carrer et al., 2010) |
| Water vapor | $H_2O$ observation (40 x 80 km – 1day) | Global Ozone Monitoring Experiment 2 Level 2 data (GOME-2 L2) | (Noël et al., 2008) |
| Surface elevation | Elevation observation (ELE) (1 m – fixed) | Digital Elevation Model (DEM) In-house database (NOAA) | (NOAA, 1988) |
| Ozone | 1-day forecast total ozone column (TOC) (1 x 1 degrees – 1 day) | Tropospheric Emission Monitoring Internet Service (TEMIS) with Assimilated Ozone Fields from GOME-2 (METOP-B) | (Eskes et al., 2003) |




















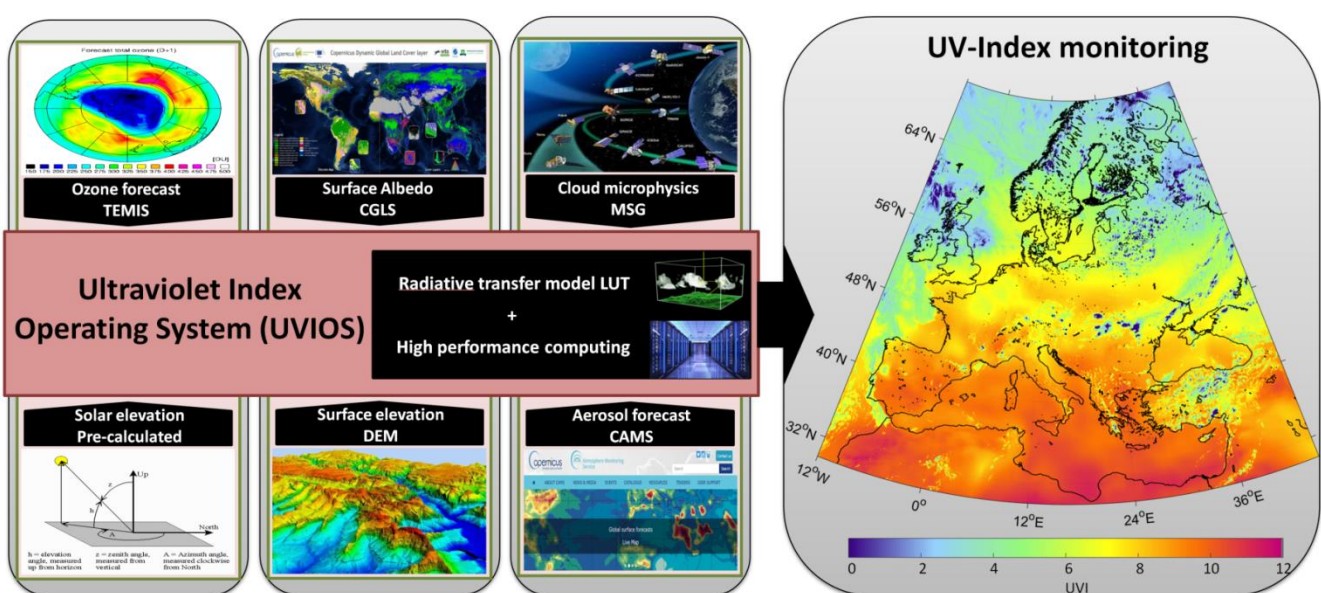


**Figure 1:** Flowchart illustration of the UVIOS modelling technique scheme. The pre-calculated effects of solar and surface elevation and albedo followed by the aerosol and ozone forecasts and the real-time cloud observations to the UVIOS solver result the spectrally weighted output of UVI for the European region.
















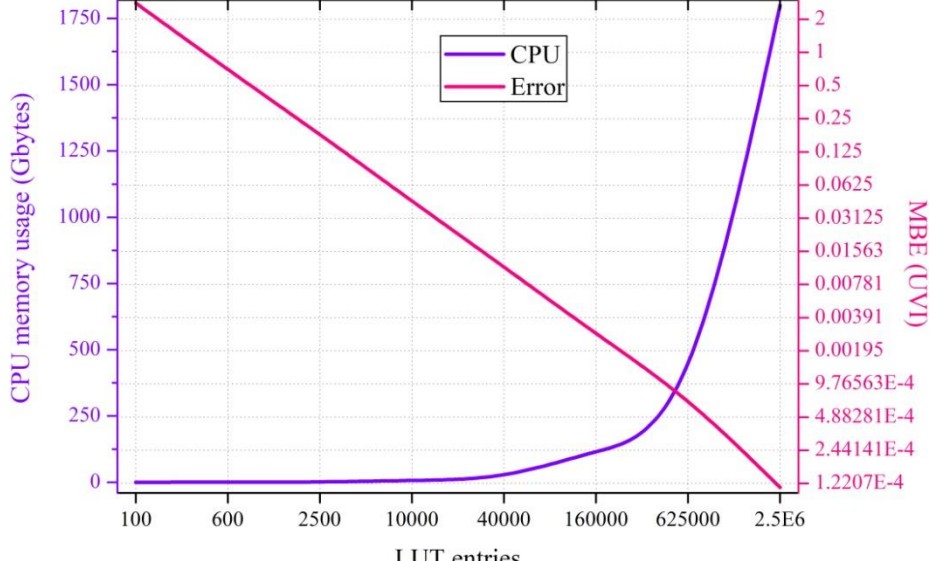


**Figure 2:** UVIOS memory usage and error statistics in terms of mean bias error (MBE) for a range of different LUT sizes.












**Figure 3:** An example of the input TOC (a), COT (b), AOD (c), SSA (d), ELE (e) and output UVI (f) maps based on the UVIOS modelling
technique applied for the 21st of June 2017 at 11:00 UTC.



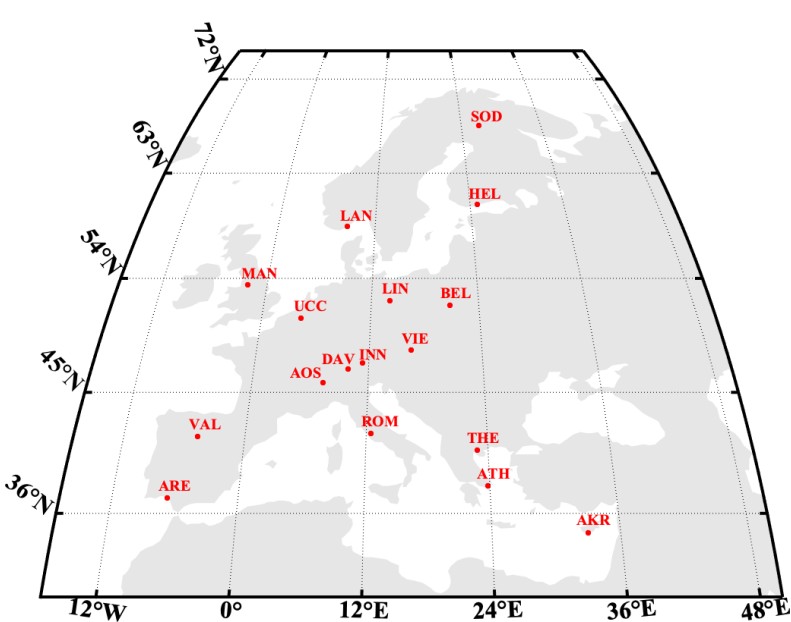

**Figure 4:** Study region and UVI ground measurement locations.










**Table 2:** Coordinates (degrees), instrument type, height (metres above sea level) and maximum UVI measured levels of the European
stations used for the comparison.

| Station | Country | Code | Latitude (ºN) | Longitude (ºE) | Instrument | Height (m.a.s.l.) | UVImax | Reference |
|---|---|---|---|---|---|---|---|---|
| Akrotiri | Cyprus | **AKR** | 34.59 | 32.99 | SL501 | 23 | 9.14 | |
| Aosta | Italy | **AOS** | 45.74 | 7.36 | Bentham DTMc300 | 570 | 9.60 | (Fountoulakis et al., 2020b) |
| El Arenosillo | Spain | **ARE** | 37.10 | -6.73 | Brewer MKIII | 52 | 9.78 | |
| Athens | Greece | **ATH** | 37.99 | 23.78 | Brewer MKIV | 180 | 10.20 | |
| Belsk | Poland | **BEL** | 51.84 | 20.79 | Brewer MKIII | 176 | 7.54 | (Czerwińska et al., 2016) |
| Davos | Switzerland | **DAV** | 46.81 | 9.84 | Brewer MKIII | 1590 | 10.57 | |
| Helsinki | Finland | **HEL** | 60.20 | 24.96 | Brewer MKIII | 48 | 5.68 | (Lakkala et al., 2008) |
| Innsbruck | Austria | **INN** | 47.26 | 11.38 | SL501 | 577 | 8.35 | (Hülsen et al., 2020) |
| Landvik | Norway | **LAN** | 58.33 | 8.52 | GUV-541 | 10 | 6.65 | (Svendby et al., 2018) |
| Lindenberg | Germany | **LIN** | 52.21 | 14.11 | Bentham DTMc300 | 127 | 8.86 | |
| Manchester | United Kingdom | **MAN** | 53.47 | -2.23 | Brewer MKII | 76 | 7.30 | (Smedley et al., 2012) |
| Rome | Italy | **ROM** | 41.90 | 12.50 | Brewer MKIV | 75 | 8.38 | |
| Sodankyla | Finland | **SOD** | 67.37 | 26.63 | Brewer MKIII | 179 | 4.51 | (Heikkilä et al., 2016; Lakkala et al., 2008) |
| Thessaloniki | Greece | **THE** | 40.63 | 22.96 | Brewer MKIII | 60 | 10.40 | (Fountoulakis et al., 2016; Garane et al., 2006) |
| Uccle | Belgium | **UCC** | 50.80 | 4.35 | Brewer MKIII | 100 | 8.99 | (De Bock et al., 2014) |
| Valladolid | Spain | **VAL** | 41.66 | -4.71 | YES | 705 | 10.32 | (Hülsen et al., 2020) |
| Vienna | Austria | **VIE** | 48.26 | 16.43 | SL501 | 153 | 8.09 | (Hülsen et al., 2020) |













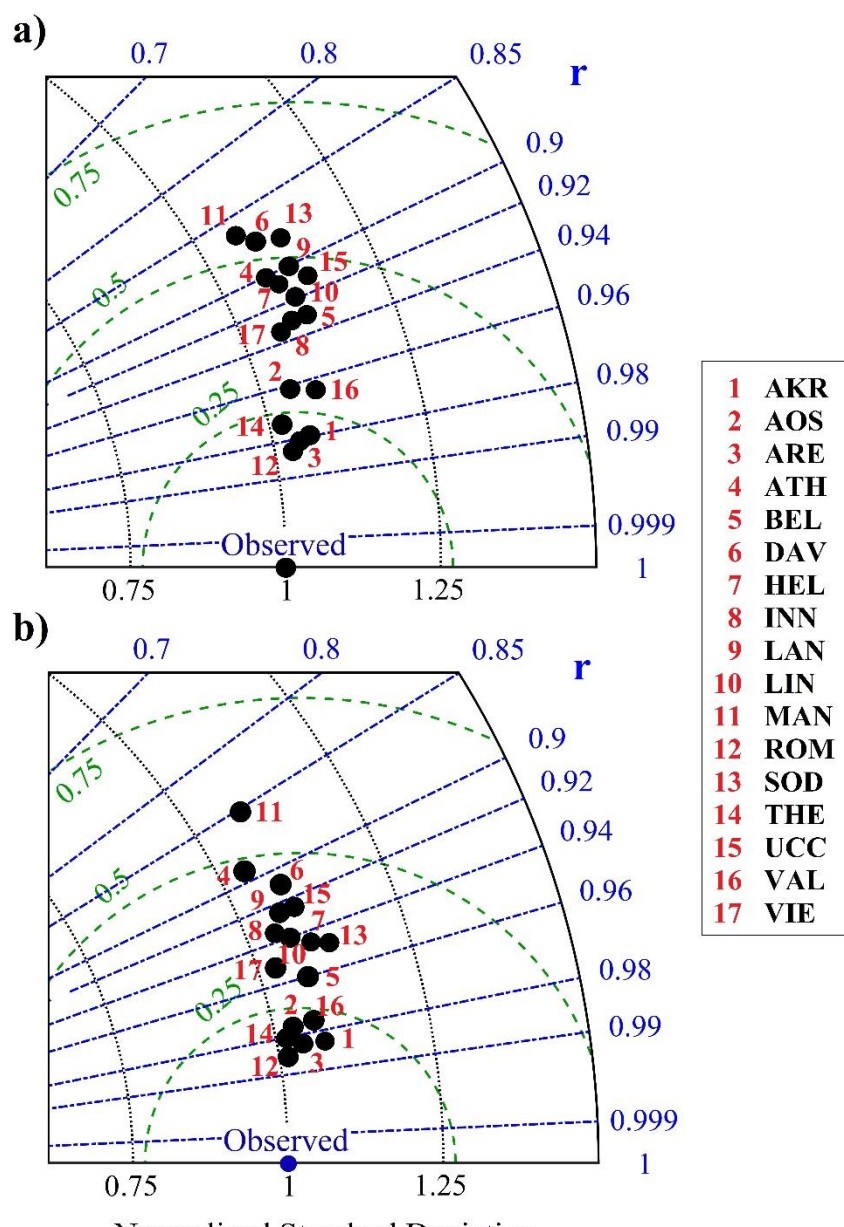



**Figure 5:** Taylor diagram for the overall UVIOS accuracy for all ground-stations under all sky (a) and clear sky (b) conditions.















**Table 3:** Absolute difference between modelled and ground based UVI measurements in terms of U0.5 and U1.0 as well as the R for all sky
and clear sky conditions.

| STATION | ALL SKY | | | CLEAR SKY | | |
|---|---|---|---|---|---|---|
| | U0.5 | U1.0 | R | U0.5 | U1.0 | R |
| AKR | 82.25 | 96.02 | 0.980 | 84.57 | 97.48 | 0.987 |
| AOS | 86.81 | 94.40 | 0.961 | 92.23 | 97.07 | 0.978 |
| ARE | 85.15 | 95.73 | 0.981 | 87.99 | 96.86 | 0.986 |
| ATH | 84.99 | 94.29 | 0.902 | 88.98 | 96.35 | 0.891 |
| BEL | 83.07 | 93.28 | 0.933 | 91.30 | 96.50 | 0.960 |
| DAV | 74.20 | 86.43 | 0.873 | 76.19 | 87.06 | 0.912 |
| HEL | 86.53 | 94.79 | 0.909 | 94.13 | 97.70 | 0.944 |
| INN | 79.96 | 92.17 | 0.932 | 87.09 | 95.23 | 0.937 |
| LAN | 84.94 | 93.46 | 0.900 | 92.34 | 96.52 | 0.925 |
| LIN | 81.58 | 91.86 | 0.919 | 90.95 | 96.31 | 0.941 |
| MAN | 77.72 | 90.44 | 0.862 | 87.85 | 94.27 | 0.852 |
| ROM | 87.69 | 96.19 | 0.985 | 89.55 | 97.00 | 0.991 |
| SOD | 90.86 | 97.26 | 0.883 | 95.69 | 98.94 | 0.947 |
| THE | 88.98 | 95.91 | 0.974 | 92.51 | 97.35 | 0.981 |
| UCC | 71.18 | 87.68 | 0.913 | 83.23 | 92.15 | 0.926 |
| VAL | 85.86 | 93.93 | 0.962 | 86.61 | 95.22 | 0.976 |
| VIE | 76.65 | 91.53 | 0.936 | 83.37 | 94.42 | 0.952 |















**Table 4:** Mean bias error of the TEMIS TOC as compared to the WOUDC ground-based measurements.

| Station | AOS | ATH | DAV | MAN | ROM | SOD | THE | UCC |
|---------|-----|-----|-----|-----|-----|-----|-----|-----|
| MBE TOC (DU) | 7.6 | -0.9 | 1.9 | 5.0 | -9.9 | -5.4 | -2.2 | 2.9 |














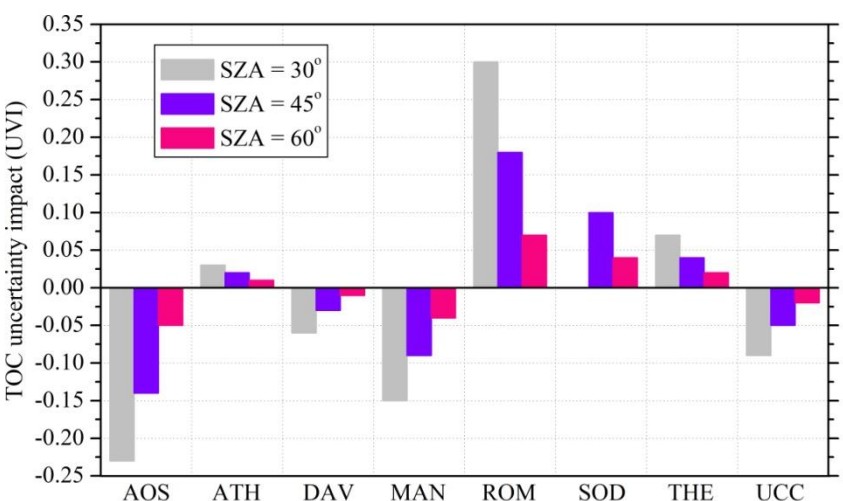


**Figure 6:** Differences of UVI derived by the UVIOS using as input the TEMIS and the Brewer TOC respectively at all stations with available data. (lower possible SOD SZA is 44 degrees).














**Table 5:** Comparison results between CAMS forecasted AOD values used as UVIOS input and AERONET ground-based AOD measurements. The AOD MBE and RMSE statistical scores are shown in absolute units, along with correlation coefficient.

| Station | AKR | ARE | ATH | DAV | HEL | LIN | ROM | SOD | THE | UCC | VAL | VIE |
|---|---|---|---|---|---|---|---|---|---|---|---|---|
| MBE | 0.037 | 0.042 | 0.030 | 0.029 | 0.062 | 0.026 | 0.017 | 0.047 | 0.008 | -0.007 | 0.024 | 0.071 |
| RMSE | 0.074 | 0.070 | 0.074 | 0.053 | 0.078 | 0.074 | 0.056 | 0.065 | 0.066 | 0.150 | 0.073 | 0.157 |
| R | 0.77 | 0.91 | 0.80 | 0.73 | 0.70 | 0.69 | 0.80 | 0.63 | 0.76 | 0.50 | 0.78 | 0.10 |



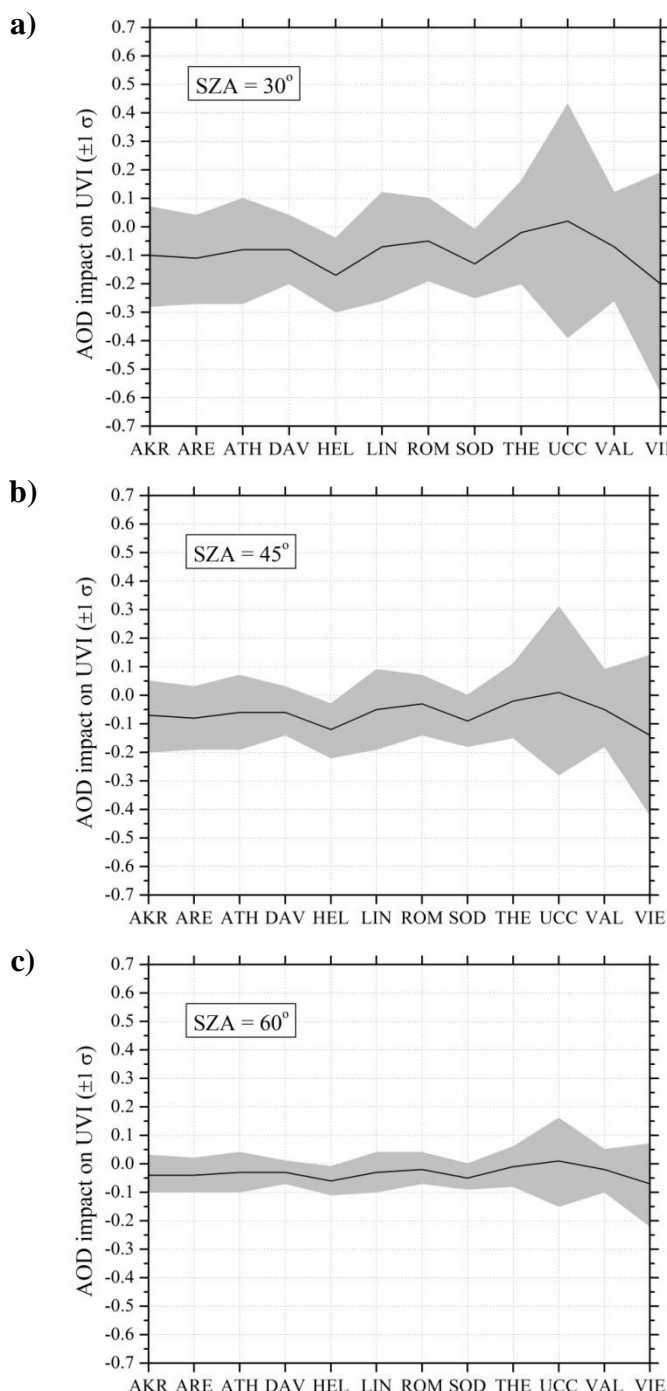

**Figure 7:** The mean bias error of the CAMS – AERONET AOD impact on UVI for all stations with available data as a function of SZA at 30 (a), 45 (b) and 60 (c) degrees together with the uncertainty range (± 1 σ).








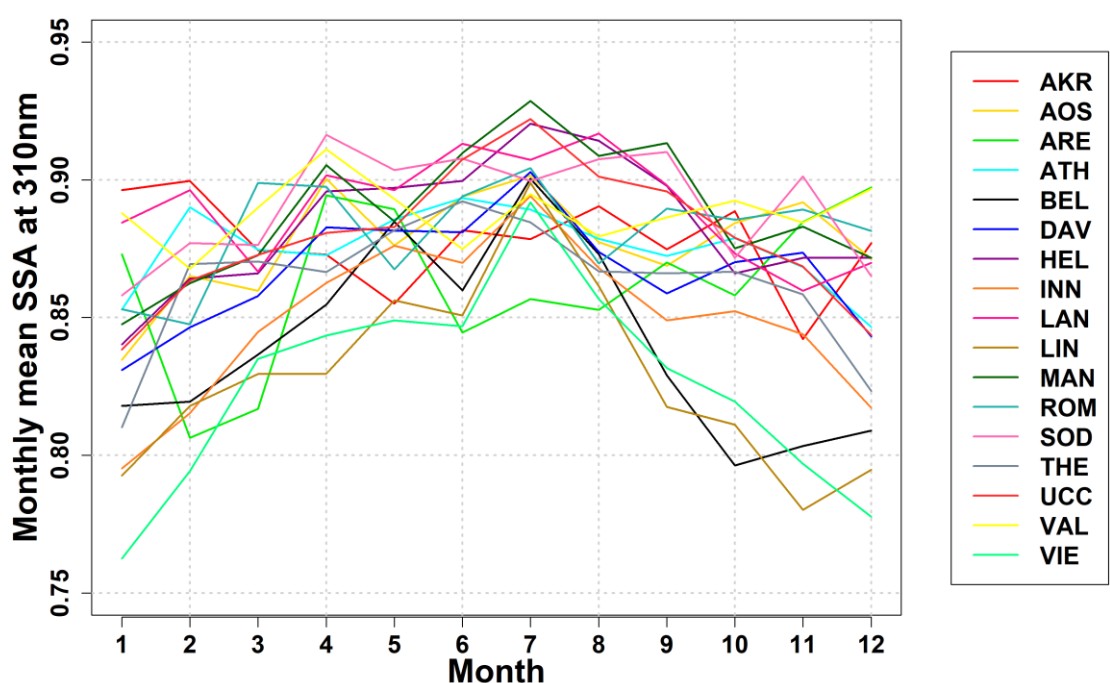


**Figure 8:** The monthly mean (i.e. 1-12 = Jan-Dec) SSA levels for all ground stations as derived by the MACv2 database.



















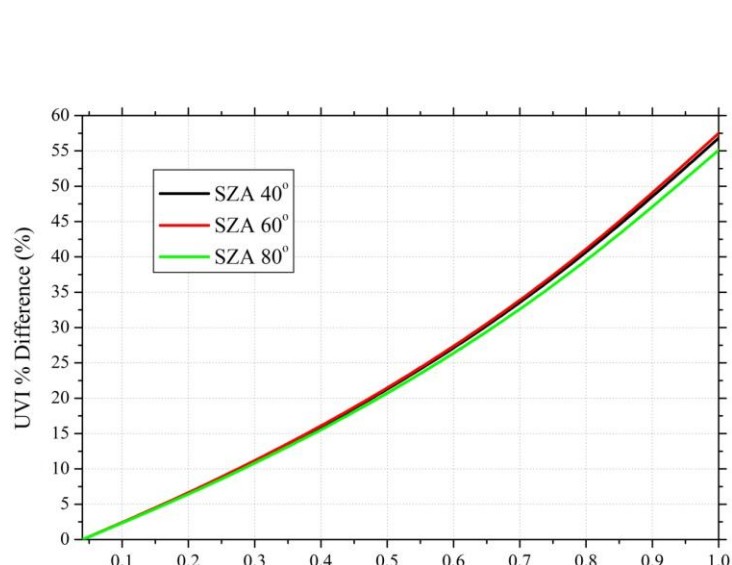


**Figure 9:** The surface albedo effect on UVI as a function of percentage difference for various SZAs.












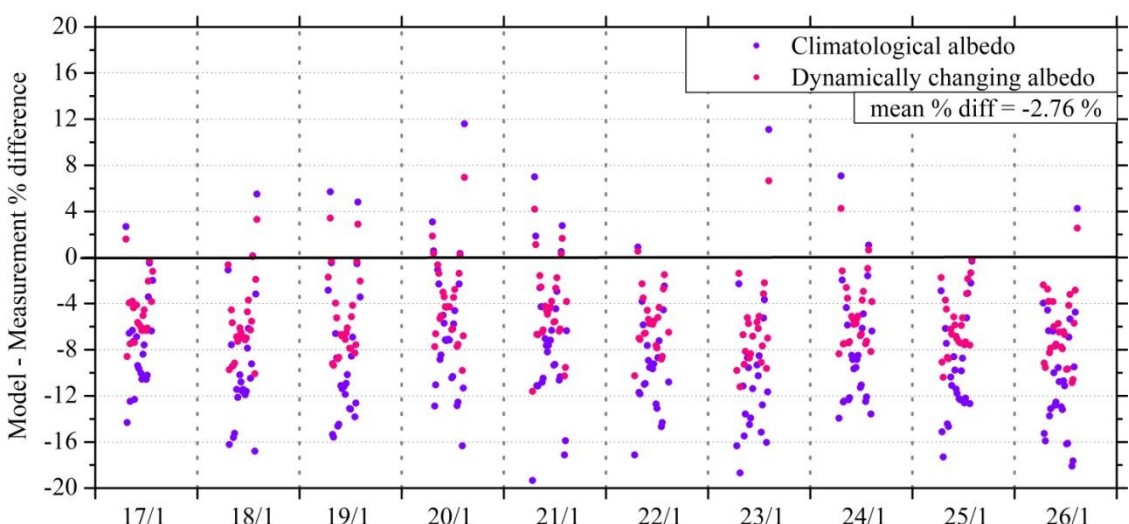

**Figure 10:** The effect of surface albedo correction on UVI for the Davos station. The climatological and the dynamically changing albedo in terms of percentage differences of modelled and ground measurements during a snow covered period (17/1 - 26/1) under clear sky conditions.



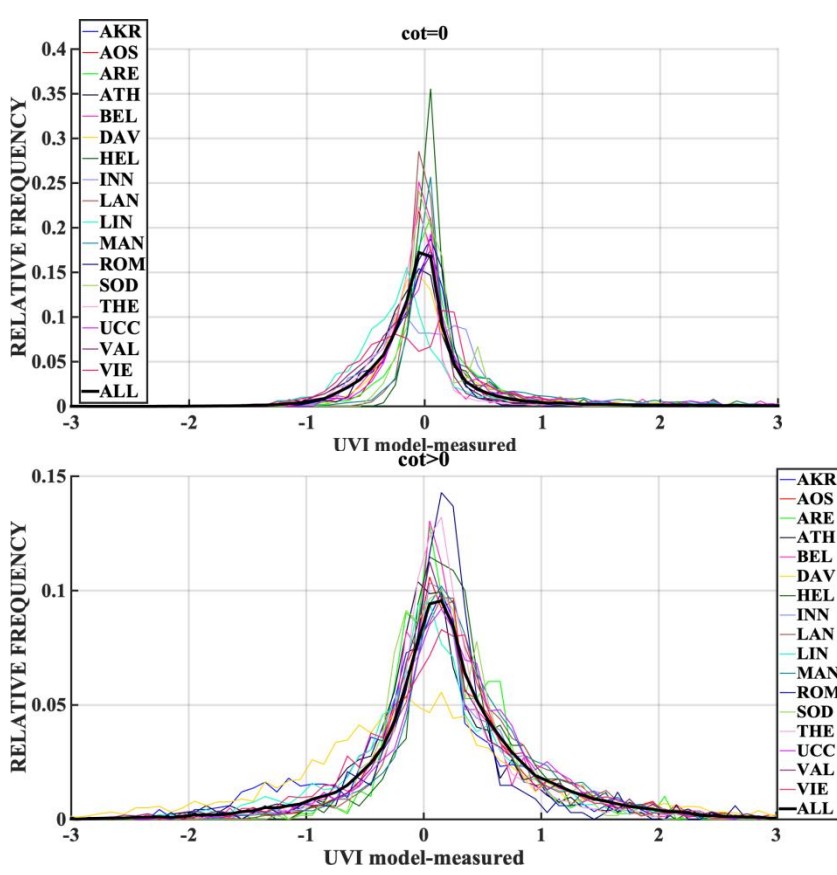

**Figure 11:** Relative frequency distribution of UVI residuals for all stations (coloured lines) and the mean (black line) for cloudless (upper plot) and cloudy (lower plot) conditions.












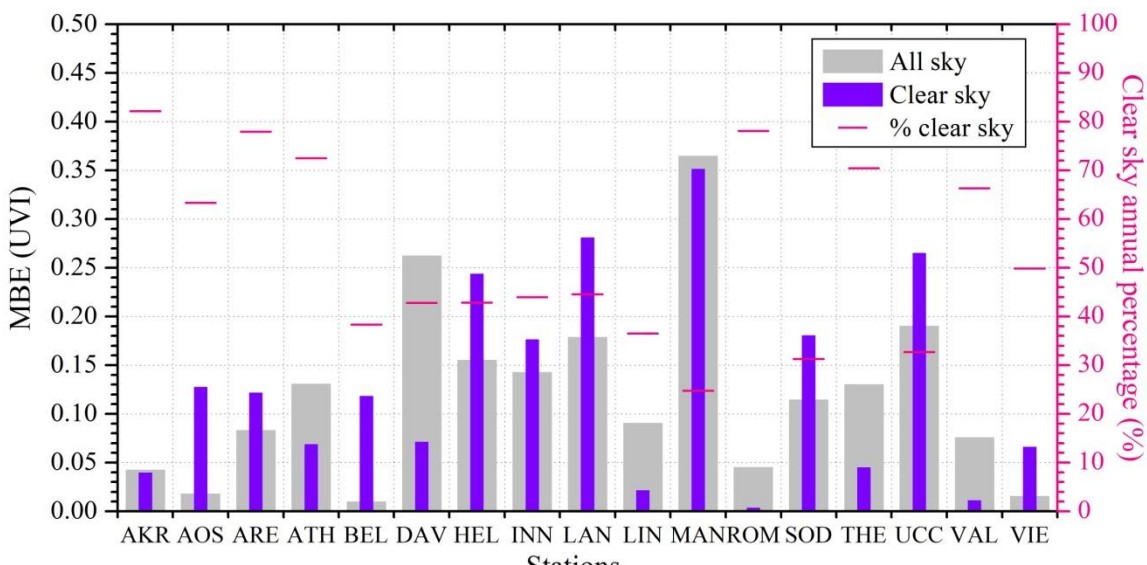


**Figure 12:** Mean bias error of the modelled UVI as compared to the ground-based measurements for all and clear sky conditions. The
percentage of clear sky data time steps was also plotted with red lines.



























**Table 6:** Percentage of data for UVIOS underestimation (A1-A3) and overestimation (B1-B3) under clear and cloudy sky conditions for
various UVI difference (modelled-ground) classes.

| Difference of UVI | < -1.0 (A1) | < -0.5 (A2) | < 0.0 (A3) | > 0.0 (B3) | > 0.5 (B2) | > 1.0 (B1) |
|---|---|---|---|---|---|---|
| % of data COT > 0 | 3.6 | 11.5 | 37.5 | 62.5 | 24.8 | 11.1 |
| % of data COT = 0 | 0.9 | 10.2 | 45.4 | 54.6 | 11.4 | 4.2 |






























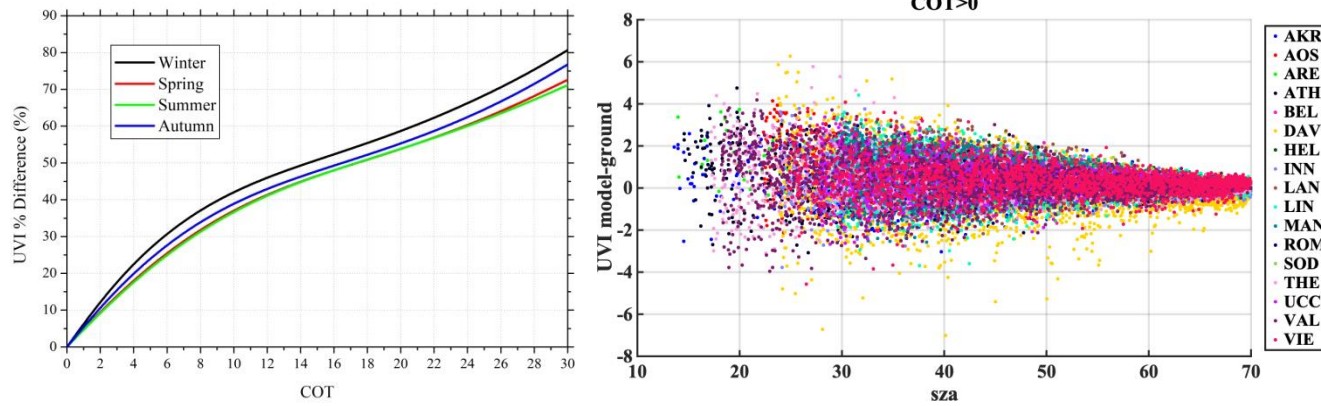

**Figure 13:** The average COT effect on UVI as a function of percentage difference for all seasons (left) and scatterplot of the UVI difference
under cloudy sky conditions for all stations (right).

















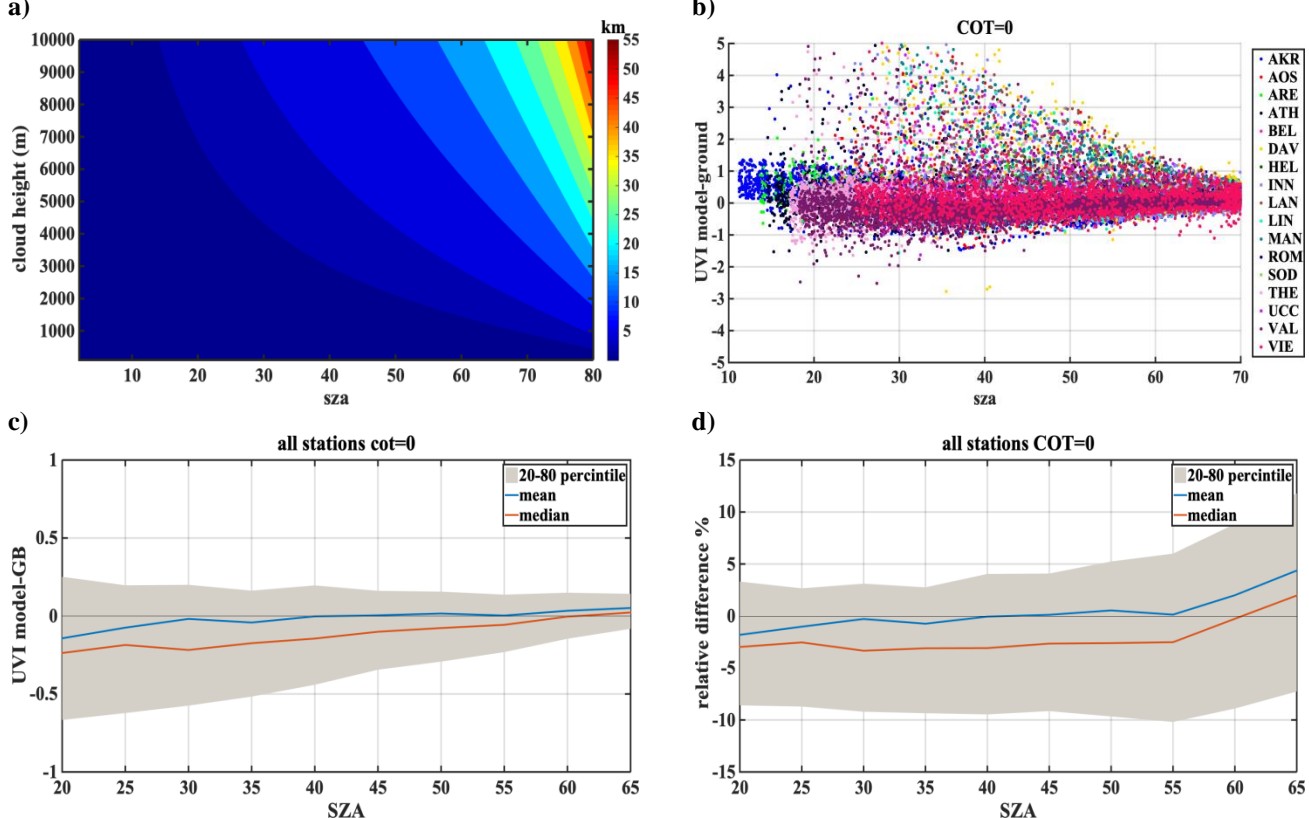


**Figure 14:** a). The surface distance of the projection of a cloud relative to the instrument in order to block the sun as a function of the cloud
height and the SZA . (b). Scatterplot of the UVI difference under clear sky conditions for all stations. c-d) UVI mean, median and 20-80
percentile differences (c) and percentage differences (d) derived by the UVIOS as compared to the ground-based measurements for clear
sky conditions as a function of SZA.















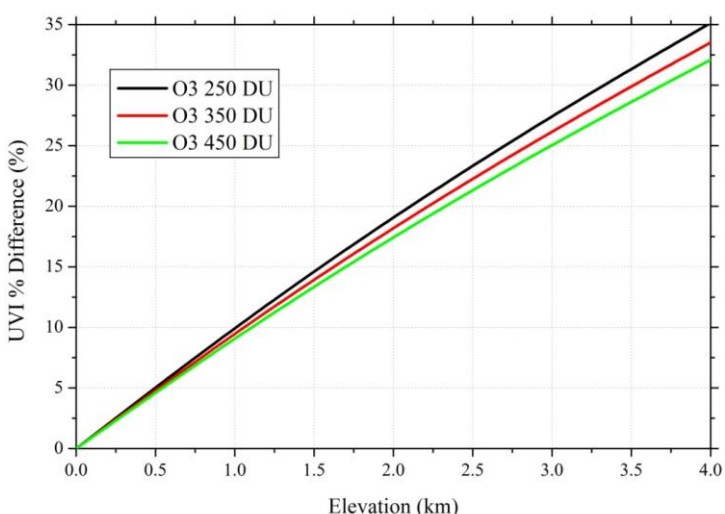


**Figure 15:** The surface elevation effect on UVI as a function of percentage difference for various total ozone columns.











**Appendix A**

The following set of Figures (A.1 – A.17) show for all stations, density scatterplots of measured and modeled UVI for all sky and clear sky conditions (upper row), normalized probability histogram of differences (middle row), and boxplot of differences (lower row) as a function of SZA, representing median (red lines), mean (blue dotted lines) 25-75 percentiles (blue boxes) and 5-95 whiskers (dotted lines).Table A.1 shows additionally the amount of data points that represent both all sky and clear sky conditions for the studied stations.

We have categorized the stations mostly based on cloud cover as Mediterranean, Central Europe, High altitude and High latitude. Each of the station has its own characteristics in terms of atmospheric conditions and parameters affecting the UVI reaching the ground. A summary of the results with possible explanation of the differences observed are shown here. The Mediterranean region includes the stations THE, ATH, AKR, ROM, VAL and ARE. Analysis of TOC showed that in most of the cases UVI mean differences are less than 0.1 in general while a negative bias between TOC and the ground measurements was seen to be highest for ROM (-9.9) that corresponds to the UVI difference of 0.3. Impact of AOD uncertainty showed the correlation coefficient between the modelled and the measured UVI values above 0.7 for most of the stations while it was as high as 0.91 for ARE. The mean bias between the modelled and measured UVI for clear sky condition was found to be less than that for the all sky condition for the stations AKR, ATH and THE that had most days of the year as cloud-free (the clear sky percentage is above 70%). The mean bias between the modelled and measured UVI for clear sky condition was more than the all sky condition for ARE even though it had mostly clear skies throughout the year. The analysis of the combined effect of the aerosol and ozone at Thessaloniki revealed that the model showed a slight underestimation with real inputs (AERONET and Brewer) while overestimations for forecasted inputs (CAMS and TEMIS). However, the coefficient of correlation was found to be as 0.989 and 0.992 for the model with forecasted and real inputs, respectively. Stations of this classification have the single scattering albedo ranging from 0.76 to 0.93, with most of them having SSA values between 0.83 to 0.93 except stations ARE and THE that had relatively smaller SSA values (0.76-0.9) and greater variability, and large MBE. AKR station comparison showed some UVIOS calculated UVI at higher levels than the GB measurements especially in low SZA's. However, GB UVI measurements seem more unrealistic than the UVIOS calculated UVI for summer local noon conditions as modeled UVIs with real AOD and TOC measurements at the area tend to agree with UVIOS outputs.

The second classification is the Central European regions including AOS, UCC, BEL, MAN, LIN, VIE and INN. The median of the absolute UVI differences between the model and the measurement for all sky condition were higher for MAN and UCC while for others it was close to zero. Larger UVI difference of -0.22 due to TOC uncertainty impact was observed for AOS which might be due to large values of UVI at higher altitude as the positive bias is highest for AOS station (7.6). The UVIOS MBE and RMSE statistical scores for analyzing AOD uncertainty impact showed a mean positive bias up to 0.071 for all the stations except UCC which is showed a mean negative bias of 0.007. The mean bias between the modelled and measured UVI for clear sky condition was more than the all sky condition for AOS even though it had mostly clear skies throughout the year. BEL, UCC and VIE showed more MBE for clear sky condition than the all sky condition as they have mostly cloudy skies



throughout the year (clear sky annual percentage less than 50%). However, stations LIN and MAN also have more MBE for
clear sky condition even though they have most days of the year as cloudy (clear sky annual percentage less than 45%).
Analysis of AOD uncertainty showed that UVI difference was highest for VIE than the other stations. The monthly values of
the single scattering albedo used in UVIOS ranged from 0.76 to 0.93 for stations AOS, UCC and MAN, with most of them
having SSA values between 0.83 to 0.93, and relatively small variability. While, the stations BEL, INN, LIN and VIE had
relatively smaller SSA values (0.76-0.9) and greater variability than the other stations and most of these stations have shown
large MBE.
The high altitude station is DAV and high latitude stations include LAN, HEL and SOD. DAV have less MBE for clear sky
condition even though they have most days of the year as cloudy (clear sky annual percentage less than 45%). DAV and MAN
show worse statistical behavior for clear sky, which is probably caused by misclassification of cloudy pixels. For DAV this
could be explained by the complex mountainous topography of the area.  Large UVI differences in SOD and HEL indicate
higher introduced uncertainties over higher latitudes. Higher aerosol levels in the atmosphere tend to lower the UVI. Highest
difference in UVI is observed for the stations HEL, SOD and VIE. Since, the aerosol level at the stations HEL and SOD is
very low this leads to higher UVI which can be the reason for th small UVI differences observed for these stations. Stations of
this classification have mostly cloudy skies throughout the year (clear sky annual percentage less than 50%) and have more
MBE for clear sky condition than the all sky condition. This might be due the fact that the clouds are not captured well at a
point station and a cloudy sky might have been considered as a clear sky. Higher UVI difference was observed for HEL and
SOD as a result of AOD uncertainty analysis which might be due to the low aerosol content of these stations due to higher
latitude that leads to higher UVI values.

Table A.1 Number of data points and clear sky data points, used in the analysis for each station.

| Station | All data | Data COT=0 |
|---|---|---|
| AKR | 6547 | 5379 |
| AOS | 5607 | 3551 |
| ARE | 1814 | 1414 |
| ATH | 4892 | 3548 |
| BEL | 1317 | 505 |
| DAV | 5635 | 2410 |
| HEL | 595 | 255 |
| INN | 4365 | 1919 |
| LAN | 7409 | 3302 |
| LIN | 3795 | 1387 |
| MAN | 7854 | 1946 |
| ROM | 1532 | 1196 |
| SOD | 860 | 269 |
| THE | 9750 | 6867 |
| UCC | 3007 | 983 |
| VAL | 9795 | 6497 |
| VIE | 4199 | 2094 |


Figure A.1



Figure A.2

Figure A.3

Figure A.4



Figure A.5



Figure A.6



Figure A.7

Figure A.8

Figure A.9

Figure A.10



Figure A.11



Figure A.12



Figure A.13



Figure A.14



Figure A.15



Figure A.16



Figure A.17
