# Peer review of "Real-time UV-Index retrieval in Europe using Earth Observation"

_Atmospheric Measurement Techniques, 2020_

## Referee Comment (RC1)

Review of the paper

Real-time UV-Index retrieval in Europe using Earth Observation based techniques and validation against ground-based measurements by Panagiotis G. Kosmopoulos et al.

The paper deals with the description of UV indices evaluation using the Earth observation system in Europe. This, so called UV-Index Operating System, or UVIOS exploits both radiative transfer models and the data available from Meteosat Second Generation and Meteorological Operational Satellite-B as well as the information available from Tropospheric Emission Monitoring Internet Service, Copernicus Atmosphere Monitoring Service and the Global Land Service.

The simulations include the account of main factors affecting UV radiation: ozone, clouds and aerosols as well as ground elevation and surface albedo with resolution of 5 km and 15 minutes.

This work is highly important "for the provision of operational early warning systems that will help raise awareness among European Union citizens of the health implications of high UVI doses" as the authors wrote.

I like the idea of this approach and its technical solution and can recommend paper for publishing. However, I have several comments, which are presented below.

Natalia Chubarova

Comments:

1. row 57. I would recommend to add the references to the numerous EEAP reports (for example, ENVIRONMENTAL EFFECTS OF OZONE DEPLETION AND ITS INTERACTIONS WITH CLIMATE CHANGE: 2014 ASSESSMENT).
2. row 73. The only mentioning UV-A as a spectral region, where the NO2 play important role seems to be misleading. See, for example, Table 7-1 from the Ozone Assessment 2006 (Chapter 7) concerning the role of NO2 and SO2 effects on erythemal irradiance. Should be clarified in the text.
3. Organic gases like formaldehyde can be also important in both UV-B and UV-A regions. I would recommend to re-write this part taking this into account.
4. row 103. The areas with extremely high positive UV trends over Northern Eurasia over the 1979-2015 period were shown recently in (Chubarova et al., 2020). (https://www.mdpi.com/2073-4433/11/1/59).
5. row 108. The reference should be given concerning the turnout point in UV trend in 2007.
6. row 145. This is not exactly so, since the method proposed by Jean Verdebout used geostationary Meteosat instruments data. This should be

accounted for in the text. (Verdebout, J., A method to generate surface UV radiation maps over Europe using GOME, METEOSAT, and ancillary geophysical data, J. Geophys. Res., [Atmos.] 105, 5049–5058, 2000. )

7. row 160. The authors should begin this part mentioning that using their approach they could combine information on input parameters from different satellite sources to provide the better quality UV estimates. I would recommend to re-write the text.

8. row 188. I do not see the information on factor of asymmetry of aerosol phase function in the list, which is one of the important aerosol parameters, necessary for model simulations. Also I do not see the cloud amount parameter in the list. I understand the difficulties with its application but this should be discussed here in the text.

9. row 198-199. The references should be provided to the internet link at least.

10. row 231. The title should be changed. Like "The description of the geophysical parameters", for example.

11. row. 248. "However, since such measurements are associated with very low UV Index (<1). Depending on different parameters ( ozone, cloud amount)." Should be proved by simulations.

12. row 261. The reference should be given to the Albedo product. The parenthesis is missed.

13. row 268. I would recommend also to add the reference to Table1 here.

14. row. 271. What is the range of overestimation?

15. row 279. Misprint ( I Note)

16. row 316. Previously you mentioned the threshold of 75 degrees for MSG COT retrievals ( row 246)? Should be clarified.

17. row after 333. I do not find the information on altitude correction. Since all other factors are analyzed here it should be also discussed here even you have the detailed analysis after.

18. row 338. I would propose to replace "while" on "and"

19. row 340-348. I would propose to re-write this part in a more compact way. This is obvious.

20. row 357. Remove the extra dot, please.

21. row 370-371. The values should be given.

22. row. 452. The reference should be given or it should be clarified that this has been obtained using model simulations provided by the authors.

23. row 458. "conditions"

24. row 468. – "conditions".

25. row 475 – change to:  and "in case"

26. row 500. Concerning the changes with altitude: there are other factors in addition to TOC, which may influence the altitude dependence like aerosol and surface albedo. Please, look for details in our paper (Chubarova et al., 2016, ACP, https://acp.copernicus.org/articles/16/11867/2016/)

27. row 507. The estimates in term of UVI should be made here to be consistent with other sections.
28. row 543. SSA? Seems to be misprint. SSA is usually used as single scattering albedo abbreviation. Here you describe the albedo effects. If you are speaking about real SSA, this should be made closer to aerosol effect discussion.
29. row 926. "result in"
30. row 1036. It would be nice to see in this Table also the RMSE and R statistics, like in Table 5.
31. Figure 6. The name of Y-axis should be changed.
32. row. 1275. (a) – is not clear.  To my understanding the angular dependence due to 3D geometry should be taken into account. Please, clarify.

---

## Community Comment (CC1)

**Example datasets from PMOD/WRC Davos, Switzerland with the double Brewer B163 and QASUME II compared with the UVIOS model**

**Julian Gröbner**

Figure 1 shows an example for a clear sky situation during the summer with nearly no cloud contamination, and the resulting good agreement between the instruments and the model. The median relative difference between UVIOS and Brewer 163 is 1.04. The mean difference is slightly larger at 1.08 with a standard deviation of 18%, due to the occasional clouds which produce significant discrepancies.

[Figure]

*Figure 1 UV Index from 4 to 6 July 2017 from measurements (Brewer B163 and QASUME II) and from the UVIOS model.*

Figure 2 shows some example conditions with scattered clouds. Due to the different spatial scale of the UVIOS model and viewing conditions, the single point comparisons can differ significantly. See for example the morning of 14 July with measured UV index around 2 and modelled UV index by UVIOS between 4 and 9.

[Figure]

*Figure 2 UV Index from 14 to 16 July 2017 from measurements (Brewer B163 and QASUME II) and from the UVIOS model.*

Figure 3 shows UV index between the model UVIOS and Brewer B163 for winter conditions, with high albedo on the ground and in the surrounding area.

[Figure]

*Figure 3 Same as before, but for winter conditions with snow on the ground. Data is from 2-10 February 2017. Only ground based measurements from Brewer B163 are available for this period.*

Another example in spring with excellent agreement during clear sky (8, 9 April) and scattered cloud conditions (5 April), but also large discrepancies (6, 7 April).

[Figure]

*Figure 4 UV Index from 5 to 10 April 2017 from measurements (Brewer B163 and QASUME II) and from the UVIOS model.*

The overall comparison between the model UVIOS and Brewer B163 for the whole period is shown in the histogram plot in Figure 5 for coincident measurements and model data within a 5 minute window. The average difference of 0.14 UV index is excellent, and the standard deviation of 0.85 UVindex even considering the periods with clouds is also remarkable. The 95% coverage of the residuals range between -1.4 to +2.5 in UVindex.

[Figure]

*Figure 5 Histogram of differences in UV index between the model UVIOS and Brewer B163 for coincident measurements in a 5 minute window.*

---

## Author Response (AR1)

**Answer to RC1**

**Author replies are in red**

The paper deals with the description of UV indices evaluation using the Earth observation system in Europe. This, so called UV-Index Operating System, or UVIOS exploits both radiative transfer models and the data available from Meteosat Second Generation and Meteorological Operational Satellite-B as well as the information available from Tropospheric Emission Monitoring Internet Service, Copernicus Atmosphere Monitoring Service and the Global Land Service. The simulations include the account of main factors affecting UV radiation: ozone, clouds and aerosols as well as ground elevation and surface albedo with resolution of 5 km and 15 minutes. This work is highly important "for the provision of operational early warning systems that will help raise awareness among European Union citizens of the health implications of high UVI doses" as the authors wrote. I like the idea of this approach and its technical solution and can recommend paper for publishing. However, I have several comments, which are presented below. Natalia Chubarova

We want to thank the reviewer for all the valuable comments and suggestions. We believe that following the proposed revisions our study was substantially upgraded.

1. row 57. I would recommend to add the references to the numerous EEAP reports (for example, ENVIRONMENTAL EFFECTS OF OZONE DEPLETION AND ITS INTERACTIONS WITH CLIMATE CHANGE: 2014 ASSESSMENT).

Reference to EAAP report has been added to the revised version of the manuscript.

2. row 73. The only mentioning UV-A as a spectral region, where the NO2 play important role seems to be misleading. See, for example, Table 7-1 from the Ozone Assessment 2006 (Chapter 7) concerning the role of NO2 and SO2 effects on erythemal irradiance. Should be clarified in the text.

We tried to clarify this sentence by expressing as follows: "Atmospheric gases play a crucial role in attenuating UV irradiance, specifically $NO_2$ is a major absorber in this spectral region (e.g. Cede et al. (2006)), while $O_3$ is the main absorber at lower (UVB) wavelengths".

3. Organic gases like formaldehyde can be also important in both UV-B and UV-A regions. I would recommend to re-write this part taking this into account.

We have added the following sentence to the updated manuscript: "Other gases that have significant absorption in the UV include $SO_2$ (Fioletov et al., 1998) and HCHO (Gratien et al., 2007), but their –usually- smaller atmospheric abundances, result in minor effects to incoming UV (with major exceptions such as volcanic incidents)."

4. row 103. The areas with extremely high positive UV trends over Northern Eurasia over the 1979-2015 period were shown recently in (Chubarova et al., 2020). ([https://www.mdpi.com/2073-4433/11/1/59](https://www.mdpi.com/2073-4433/11/1/59)).

We have added the following sentence: "Chubarova et al. (2020) found a long term increase of 3% per decade in UV at Northern Eurasia, for the 1979-2015 period". Thank you for the excellent suggestion.

5. row 108. The reference should be given concerning the turnout point in UV trend in 2007.

The reference to Zerefos et al. (2012) has been added.

6. row 145. This is not exactly so, since the method proposed by Jean Verdebout used geostationary Meteosat instruments data. This should beaccounted for in the text. (Verdebout, J., A method to generate surface UV radiation maps over Europe using GOME, METEOSAT, and ancillary geophysical data, J. Geophys. Res., [Atmos.] 105, 5049–5058, 2000. )

The correct reference of Verdebout is provided in the new version of the manuscript. Thank you for the careful reading.

7. row 160. The authors should begin this part mentioning that using their approach they could combine information on input parameters from different satellite sources to provide the better quality UV estimates. I would recommend to re-write the text.

In the revised manuscript we provide a completely updated version of this paragraph. Now we believe that the presentation of this part, as an introduction to the UVIOS system, has been substantially improved. Thank you for the valuable comments and suggestions.

8. row 188. I do not see the information on factor of asymmetry of aerosol phase function in the list, which is one of the important aerosol parameters, necessary for model simulations. Also I do not see the cloud amount parameter in the list. I understand the difficulties with its application but this should be discussed here in the text.

The asymmetry factor was also retrieved from the MACCv2 climatology (Kinne, 2019). The specific information has been added to the manuscript. Analytical discussion for clouds is already provided in lines 192-196, as well as a reference wherein more information can be found (Taylor et al., 2016). We do not believe that further discussion for clouds is necessary.

9. row 198-199. The references should be provided to the internet link at least.

References have been added.

10. row 231. The title should be changed. Like "The description of the geophysical parameters", for example.

We agree with the proposed new title which describes better this subsection. It was changed in the updated version of the manuscript.

11. row. 248. "However, since such measurements are associated with very low UV Index (<1). Depending on different parameters ( ozone, cloud amount)." Should be proved by simulations.

This part was revised accordingly in order to be clearer and indicate the cloud information availability and reliability issues under high solar zenith angles (>70 deg). Concerning the fact that under such high solar zenith angles the accompanied UVI levels are <1, below we plotted indicatively the UVI data of Thessaloniki for solar zenith angles >70 degrees. The new version of this paragraph changed as follows: "UVIOS calculations at high solar zenith angles (>70 deg) are retrieved assuming cloudless skies since the MSG COT product is not available in these conditions, facing reliability issues (Kato and Marshak, 2009). This has an effect on the quality of the UVIOS overall performance at high solar zenith angles, where there is no cloud information as input to the model in order to quantify the consequent impact on UVI. However, such measurements under high solar zenith angles are accompanied with very low UVI levels (<1) both in the performed RTM simulations and in the ground-based measurements. This inconsistency, even if does not affect UVIOS UVI results associated with dangerous effects on human health, nevertheless it is still affected by the rest of input parameters (i.e. ozone, aerosol etc) mitigating the UVIOS uncertainty in the absence of cloud information under such high solar zenith angles. There is more discussion in the next section on how we use these data for the UVIOS validation."

[Figure]

The retrieved UVI data of Thessaloniki for solar zenith angles >70 degrees.

12. row 261. The reference should be given to the Albedo product. The parenthesis is missed.

Done.

13. row 268. I would recommend also to add the reference to Table1 here.

The reference to Table 1 was added.

14. row. 271. What is the range of overestimation?

The range overestimation during 2017 is from 0 to 0.4 in terms of modified normalized mean bias (see at Eskes at al., 2015), with the same range of values over the study region (Europe). We revised this part accordingly in order to include the above information.

15. row 279. Misprint ( I Note)

Thank you for the careful reading. It was corrected in the updated version.

16. row 316. Previously you mentioned the threshold of 75 degrees for MSG COT retrievals ( row 246)? Should be clarified.

The 75 degrees in row 246 was a misprint. It was corrected in the updated version of the manuscript.

17. row after 333. I do not find the information on altitude correction. Since all other factors are analyzed here it should be also discussed here even you have the detailed analysis after.

The suggested information was added in the manuscript as follows: "Although UVIOS simulations were corrected for changing UVI with respect to altitude (see Section 3.2.3), the correction cannot be perfect for higher altitude stations. The reason is that it is not possible to take into account all different factors (aerosol load and properties, atmospheric pressure, surface albedo) (e.g., Blumthaler et al., 1997; Chubarova et al., 2016) which affect the change of UVI with altitude."

18. row 338. I would propose to replace "while" on "and"

This paragraph has been completely re-written.

19. row 340-348. I would propose to re-write this part in a more compact way. This is obvious.

The specific section has been re-written in a clearer and more concise way.

20. row 357. Remove the extra dot, please.

Removed.

21. row 370-371. The values should be given.

The values were added in the document.

22. row. 452. The reference should be given or it should be clarified that this has been obtained using model simulations provided by the authors.

The provided numbers are based on the UVIOS simulations. A clarification sentence has been added.

23. row 458. "conditions"

Corrected.

24. row 468. – "conditions".

Corrected.

25. row 475 – change to: and "in case"

Corrected, thank you for the careful reading.

26. row 500. Concerning the changes with altitude: there are other factors in addition to TOC, which may influence the altitude dependence like aerosol and surface albedo. Please, look for details in our paper (Chubarova et al., 2016, ACP, https://acp.copernicus.org/articles/16/11867/2016/)

Related discussion has been added in the revised Section 2.4.

27. row 507. The estimates in term of UVI should be made here to be consistent with other sections.

This part was revised accordingly in order to include the surface elevation correction and effect in terms of UVI. The updated text is the following: "Indicatively, the average maximum surface elevation correction in terms of UVI for the DAV station (due to UVIOS input deviation from to actual elevation) was of the order of 1.6 (15%), while for INN and AOS it was 0.5 and 0.6 respectively (6%) and for the VAL station close to 0.8 (8%)."

28. row 543. SSA? Seems to be misprint. SSA is usually used as single scattering albedo abbreviation. Here you describe the albedo effects. If you are speaking about real SSA, this should be made closer to aerosol effect discussion.

This part of the manuscript was not clearly written. The discussion was indeed for the single scattering albedo (SSA), and now it has been moved directly after the discussion for the AOD uncertainties.

29. row 926. "result in"

Corrected.

30. row 1036. It would be nice to see in this Table also the RMSE and R statistics, like in Table 5.

In the updated manuscript, the RMSE and r were added in Table 4, similar to Table 5.

31. Figure 6. The name of Y-axis should be changed.

The name of Y-axis in Fig. 6 was changed into "UVI difference (TEMIS – Brewer)".

32. row. 1275. (a) – is not clear. To my understanding the angular dependence due to 3D geometry should be taken into account. Please, clarify.

In the revised manuscript, this part has been explained more clearly in the text and in the Figure 15 (a) caption. With this plot we simulated the shadow volume at the surface level of a cloud relative to the SEVIRI angle view, as a function of various cloud heights and solar zenith angles. To this direction we tried to highlight in a simple way the observed ray tracing by SEVIRI in the presence of clouds, and hence, the angular dependence due to the 3D geometry. We want to thank the reviewer for this apposite remark and the overall analytical review that upgraded our manuscript. Thank you.

**Answer to RC2**

**Author replies are in red**

The manuscript evaluates the UV index generated from the UV- Index Operating System (UVIOS) and also conducted sensitivity tests to quantify the uncertainty caused by the different input parameters. The system is important for providing early warning, which will benefit the general public. The evaluation of the UV estimation against ground observation is thorough. However, the presentation of the manuscript can be improved, and I have provided some detailed comments as follows. Lastly, English needs to be improved. There are grammar errors in some sentences.

We want to thank the reviewer for the proposed comments, suggestions and corrections that were incorporated in the revised version of the manuscript.

1. The manuscript mentions that the UVIOS system provides both now-casting and forecasting UV index products. It is not clear how different they are and it would be helpful to provide some information on these two products.

Thank you for giving us the opportunity to explain in more detail the ways that UVIOS is able to operate. Since this system can produce massive UVI outputs of the order of 1.5 million simulations in less than 5 minutes following the proposed computing architecture (section 2.1.2), this means that it can be used for both operational applications and real-time estimations. The exact use of UVIOS depends only on the available input data sources. For this study both nowcasts (clouds) and forecasts (ozone, aerosol) were used as inputs into UVIOS. The nowcasts represent the continuous monitoring dimension (i.e. what is happening now) in terms of cloud microphysics data every 15 minutes retrieved in real-time by the geostationary satellite Meteosat Second Generation (MSG). The forecasts represent the future estimations (day ahead in our study) of aerosol optical properties and total ozone column based on deterministic approaches (ECMWF) and assimilated satellite data for better accuracy. As a result, UVIOS under cloudless conditions operates as a forecast system since it uses forecasted inputs and provides the clear-sky UVI forecasts operationally. By adding the nowcast cloud information as input to UVIOS (i.e. all sky conditions), the whole procedure will follow the time steps of MSG cloud microphysics data collocated and synchronized with the forecast data. So, following the proposed operation method of this study, the UVIOS can be used as a UVI forecast system for cloudless conditions or as a UVI nowcast system for all sky conditions. A future goal is to compare the UVIOS accuracy under cloudy conditions by using, (i) the current MSG cloud information (5 km, 15 min), (ii) the ECMWF forecast cloud information (4 km, 1 hour) and (iii) the forthcoming Meteosat Third Generation (MTG) cloud information (500m, 5 min), in order to quantify the uncertainties of the forecasted cloud data as compared to the satellite observations, as well as the overall improvement of the MTG data compared to the MSG due to the MTG's higher resolution.

The above text has been added in the revised version of the manuscript in order to clarify the UVIOS operating principles and the dependence on the input data sources.

2. Line 298-302 talks about the different stations. It can be moved to the next paragraph where it focuses on describing the geographic and climate information about each station.

These lines have been moved to the next paragraph. Now the revised paragraph includes all the geographic and climate information for all stations. Thank you for the suggestion, which indeed, improved the presentation of this part.

3. I suggest changing title 3.1 to Overall performance of the UVIOS system.

The title of subsection 3.1 has been changed following the reviewer's suggestion.

4. Line 368: how did you decide the criteria for low, moderate and high UVI differences?

There was a typo in the document. The three categories are: low (difference<0.5), moderate (0.5 – 1), and high (differences>1). We used the above categorization for the following reasons:
- Differences below 0.5 are of similar magnitude with the measurement uncertainties and are thus considered low.
- Differences larger than 1 correspond to significant differences in the health effects of the different UVIs, and different precaution measures. For example, a UVI of 5 is considered moderate while a UVI of 6 is considered high.
- Finally, differences between 0.5 and 1 are not that significant regarding the health effects of the different UVIs, and are thus categorized as moderate.

5. For Figure 4, it would be nice to show a map of the annual UV index values from the UVIOS system overlaid with ground observational data.

We totally agree that an annual UV index map produced by UVIOS would be of great interest. To this direction a sequent study is already planned in which a large-scale historical database of high resolution UV index will be developed using all the available climatological and historical input data sources followed by a dedicated online and open access application (web-service). This procedure will require huge computing and storage resources in order to implement (e.g. for 15 years covering Europe and North Africa). Indicatively, just for one-year collocated cloud, aerosol, ozone and surface albedo data for the same region used in this study (2 million pixels), the total amount will be more than 70 billion UVIOS simulations and 25 Tb of input and output data production and use. We want to thank the reviewer for the nice point of view concerning the climatological dimension of UVIOS and we believe that the planned long-term database development will be useful as well for the UV community.

6. Besides Figure 5, it would be helpful to have a scatter plot here to better show the overall performance for all the stations together.

A scatterplot (new Fig. 5) including all stations in order to show the overall model performance is provided in the revised version of the manuscript, additionally to the Taylor diagram (now Fig. 6).

7. Table 3 is hard to read. Figure 11 can be used as a complement. I would suggest showing a histogram distribution of the errors for all the stations together rather than showing each station on one plot with different lines. Histogram for individual station can also be shown as supplementary plots.

We want to thank the reviewer for the helpful suggestions. In the revised version of the manuscript a more explanatory caption of Table 3 is provided for the presented quantities (i.e. U0.5 and U1.0 represent the percentages (%) of data that are within 0.5 and 1 UVI of difference, respectively). Additionally, separating lines were added for better clarity between the station column and the all sky and clear sky statistics. Concerning Fig. 11, we have updated this plot in order the distribution of the whole dataset to be clearer. The distribution of all data is now thicker and station by station lines are thinner. We think that now the information and the visualization of this plot were updated followed by better clarity.

8. Figure 9 and Figure 15 can be put together into one figure.

Figures 9 and 15 were merged into one plot (new Fig. 10) and the related texts were included in the same subsection (3.2.3).

9. Line 84: what is "0 to 8/8"?

It is referring to cloudiness oktas, we have rephrased accordingly now in order to be more clear, i.e. "the change from 0 to 8 oktas for cloud coverage".

10. Line 85-88: This sentence has some grammar errors, please check and reorganize the sentence.

The sentence has been restated as follows: "Although, the transmittance of clouds does not vary significantly with wavelength, some studies (Mayer et al., 1998; Seckmeyer et al., 1996) have found that the diffuse component of the surface UVR is affected by clouds in a spectrally dependent way, due to more efficient scattering and absorption of shorter UV wavelengths, in case of large air masses".

11. Line 91: It is not necessary. Or what is the main conclusion of Calbo et al. (2005)?

This sentence has been deleted in the revised version of the manuscript, since we are not discussing details of this subject.

12. Line 103: what does it mean by in contrast to ozone"? Is ozone column not important for the trend?

This phrase has been eliminated because it is confusing. Our purpose was to highlight the importance of other parameters (clouds, aerosols) to UVR trends, since the influence of ozone is described in detail earlier in the manuscript.

13. Line 160: change next section to Section xx.

We thank the reviewer for the useful suggestions in order to succeed a better presentation of the manuscript. We followed both reviewer's suggestions concerning the structure and presentation and ended up into the provided version.

14. Line 271: change Aeronet to AERONET.

Corrected.

15. Line 339: which section is it for the cloud effect section?

The cloud effect section in the revised version of the manuscript is the 3.2.4.

The authors of this study appreciate the valuable corrections, comments and suggestions of the reviewer and believe that the revised version is substantially upgraded. Thank you.

---

## Author Response (AR2)

**Answer to Anonymous Referee #1**
**Author replies are in red**

Specific Comments

I appreciate the authors' efforts to address my previous comments and the manuscript has indeed improved. I still have some comments as follows:

1) I think the title could be revised to better reflect the work and make the paper more complete. Maybe it can be something like this: Real-time UV-Index retrieval in Europe using Earth Observation based techniques: system description, quality assessment and future plan

This way, it can help better organize the paper structure. I think this paper serves as the overview paper of the system, which is just the beginning. With that, I would like to see an additional section to talk about some of the future plans with the system, such as how the data can be accessed and used etc. Also, it might be good to break section 2 into two sections: once section can be "section 2: system description which focuses on the detailed description of the component of the system such as the input data and the nowcasting products; another section "section 3: ground measurements and evaluation methodology. Then section 4: "results and future plan" can be together or separate.

We thank the reviewer for the constructive suggestions that indeed upgraded the manuscript and the presentation quality. In the revised version we changed the title into "Real-time UV-Index retrieval in Europe using Earth Observation based techniques: system description and quality assessment". The future plan part was added in the last paragraph of the updated "Section 5: Conclusions and future plans". Section 2 was also divided into the following two sections: "Section 2: The UV Index operating system (UVIOS)" including the subsections "2.1: System description", "2.2: Real-time processing concept" and "2.3: Input data description". Then, we placed the "Section 3: Ground measurements and evaluation methodology" which consists of the subsections "3.1: Ground-based measurements" and "3.2: Evaluation methodology". Finally, the Results section was adjusted to "Section 4: Results" and the final section was renamed as "Section 5: Conclusions and future plans" in order to provide some brief information about the future plans with the system.

2) The introduction is still long and I have provided several comments here to make it more concise and readable:

- This work is not focusing on the trend analysis, line 104-123 discusses a lot of studies of trend analysis. I understand that the point is to show the importance of continuous monitoring of UV index. I would suggest cutting this short to keep only one paragraph and indicate the main points without the need to describe each study.

The new paragraph is significantly shorter with less studies discussed in detail.

-Line 142-167: it is a very long paragraph and hard to read. I would suggest breaking this down into two paragraphs

The part discussing the comparison of satellite to ground based retrievals has been separated to a new paragraph.

-Line 168-193: again, I appreciate the authors to explain the nowcasting and forecasting products, which can be overwhelming here. I would suggest moving this bulk of information to Section 2 where this system is introduced.

This part was moved to the new Section 2.2.

3) Line 56, 59: It would be helpful to add the wavelength range for the UV radiation and UVB radiation, respectively.

Wavelength ranges have been added.

4) Line 72: In this spectral region, what is "this spectral region"?

Changed to UV.

5) Figure 5 caption: please describe the color bar in the caption.

The caption was corrected as to describe this type of plot, i.e. density scatter plot in which a pattern of shaded squares represents the counts of the points falling in each square. The aforementioned brief description was added in the Fig. 5 discussion.

6) Table A.1 is not necessary. It is better to put the # of data points on the respective figure so that the information would be available right away to readers when they read the figure without going back to the table.

Corrected.

7) I would suggest the authors to do a round of thorough proofreading of the manuscript.

The proofreading of the manuscript was performed by all authors individually.

Corrections:
Line 78-80: this sentence needs to be reorganized: " as well as single scattering albedo (SSA) " there is confusing

We have reorganized in two sentences and now it reads as:
"Aerosol optical depth (AOD) that quantifies the attenuation of the direct solar beam by aerosols is a parameter varying with wavelength. Single scattering albedo (SSA), which determines the scattering ratio to total extinction, is also a spectrally variant parameter"

Line 80: what do you mean by "incoming UV irradiance measurements", surface UV irradiance measurements?

Has been changed to surface.

Line 81-83: This sentence to show the conclusion from these literature reviews is a little confusing. They show the importance of using SSA in the UV spectral region, what did they use the SSA for? What did they study?

The sentence now reads as:
"Finally, a number of studies have highlighted the importance of using representative SSA in the UV spectral region, instead of interpolating SSA at visible wavelengths to the UV, or directly using SSA at visible wavelengths, options that systematically overestimate UV irradiance (Corr et al., 2009; Fountoulakis et al., 2019; Kazadzis et al., 2016; Mok et al., 2018; Raptis et al., 2018)."

Line 375: change "which is followed by" to "which shows"

Corrected.

Line 375: the correlation coefficient is "0.944", there is no need to keep 3 decimal places as other places have used two decimal places, it is better to be consistent. Again, the decimal places on the figures in the Appendix are like "R = 0.98005", please double check.

Corrected to two decimal places throughout the manuscript including the Appendix figures.

Line 377: "For both, clear sky and all sky", remove the ","

Corrected.

Line 571: add "," between "to date" and "polar orbiting"

Corrected.

Line 577: change "The mean differences TEMIS TOC" to "The mean differences between TEMIS TOC…"

Corrected.

Following the proposed correction, we believe that the manuscript in now complete. We want to thank the reviewer once again for his support and for making this study even more clear and reader friendly.